# Glance and Focus: Memory Prompting for Multi-Event Video Question Answering

**Ziyi Bai**[1,2]**, Ruiping Wang**[1,2]**, Xilin Chen**[1,2]

[1]Key Laboratory of Intelligent Information Processing of Chinese Academy of Sciences (CAS),
Institute of Computing Technology, CAS, Beijing, 100190, China
[2]University of Chinese Academy of Sciences, Beijing, 100049, China
*ziyi.bai@vipl.ict.ac.cn*, *{wangruiping, xlchen}@ict.ac.cn*

## Abstract

Video Question Answering (VideoQA) has emerged as a vital tool to evaluate agents' ability to understand human daily behaviors. Despite the recent success of large vision language models in many multi-modal tasks, complex situation reasoning over videos involving multiple human-object interaction events still remains challenging. In contrast, humans can easily tackle it by using a series of episode memories as *anchors* to quickly locate question-related key moments for reasoning. To mimic this effective reasoning strategy, we propose the Glance-Focus model. One simple way is to apply an action detection model to predict a set of actions as key memories. However, these actions within a closed set vocabulary are hard to generalize to various video domains. Instead of that, we train an Encoder-Decoder to generate a set of dynamic event memories at the glancing stage. Apart from using supervised bipartite matching to obtain the event memories, we further design an unsupervised memory generation method to get rid of dependence on event annotations. Next, at the focusing stage, these event memories act as a bridge to establish the correlation between the questions with high-level event concepts and low-level lengthy video content. Given the question, the model first focuses on the generated key event memory, then focuses on the most relevant moment for reasoning through our designed multi-level cross-attention mechanism. We conduct extensive experiments on four Multi-Event VideoQA benchmarks including STAR, EgoTaskQA, AGQA, and NExT-QA. Our proposed model achieves state-of-the-art results, surpassing current large models in various challenging reasoning tasks. The code and models are available at
`https://github.com/ByZ0e/Glance-Focus`.

## 1 Introduction

For intelligent agent that can assist humans in completing daily-life tasks, complex event reasoning ability is particularly important. Multiple Event Video Question Answering (Multi-Event VideoQA) task [14, 42, 16, 43] has recently been proposed. These benchmarks require agents to reason about complex videos depicting multiple human-object interactions (i.e., events or situations). Furthermore, a wide scope of event-centric questions are involved[14], ranging from event semantic descriptions to event sequencing and dependencies to comprehensively evaluate an agent's ability to understand human behavior.

Recently, the emerged large vision-language models [1, 3, 8, 25, 47, 50] have demonstrated impressive generalization capabilities in VideoQA tasks. By leveraging pre-training on large-scale cross-modal pairs, these models can establish correlations between questions and videos to find answers effectively. Such approaches proves effective for short videos containing single or few events with one-step

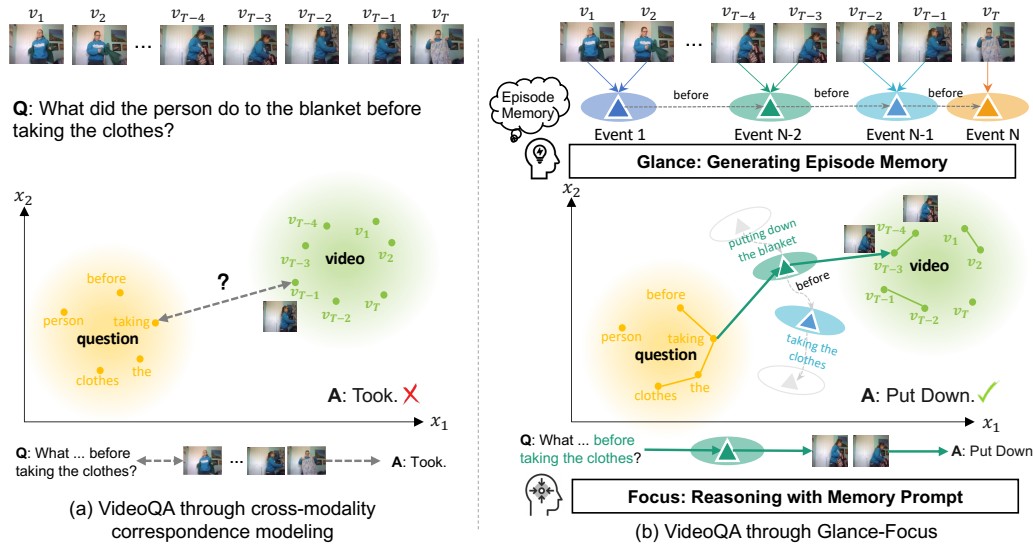

Figure 1: Comparison of different VideoQA solutions. The cross-modal common feature space is visualized. (a) Cross-modal correspondence modeling. Weak correspondence between the question with high-level event concepts and low-level lengthy video leads to a wrong answer. (b) Human-like Glance-Focus Strategy. Glancing stage: a set of episode memories are generated. Focusing stage: memory prompts are used as anchors to quickly locate the keyframes for reasoning.

questions like "*What is someone doing?*". However, Multi-Event VideoQA poses distinct challenge. As shown in Figure 1(a), due to the longer video duration and more events involved, it is challenging for the model to effectively find the visual clues from the lengthy video content. Additionally, the event-centric questions often include high-level event concepts while the semantics of video frames are relatively low-level, making it even more difficult to directly locate key moments related to the question.

This is very similar to the difficulties we humans encounter when addressing reading comprehension tasks, where we must quickly extract question-related context from long paragraphs. To ensure fast and effective reasoning, it is common for humans to employ a strategy of first glancing through the story, summarizing the storyline in our mind, and then using this episode memory to quickly focus on the relevant context when answering questions.

We mimic such a reasoning strategy to tackle the challenging Multi-Event VideoQA task with our proposed Glance-Focus model. It comprises two stages: glancing and focusing. The purpose of the glancing stage is to generate a set of episode memories that can efficiently summarize the multiple events occurring in the video. A naive way is to directly train an action detection model to obtain a set of actions. However, this would result in unnecessary computation, and most importantly, it may not be practical for encapsulating all possible actions in the real world using closed vocabularies.

Instead, we adopt a more flexible approach that automatically obtains a set of dynamic event memories using a Transformer Encoder-Decoder architecture [39]. For datasets with closed event vocabularies, we employ bipartite matching between the extracted memories and ground truth labels to explicitly obtain event memories. Apart from supervised event matching approach, we additionally design an unsupervised event generation way to get rid of the event-level annotations of the video. We impose individual certainty and global diversity constraints during event generation to ensure they can effectively summarize the video content.

Finally, since these generated memories are still independent events, we reorganize them into complete event memory prompts by arranging them in sequence order with temporal information remaining. As shown in Figure 1(b), since these memory prompts have high-level event semantics, they play an anchor role in the multi-modal common feature space to help quickly locate the related video content according to the question. Such as the example in Figure 1(b), we design a multi-level cross

attention mechanism to first focus on the event "putting down the blanket" before "taking the clothes" according to the question, then focus on the related video content for answer prediction.

We conduct experiments on various recently released Multi-Event VideoQA datasets, STAR [42], AGQA [14], EgoTaskQA[16] and NExT-QA[43]. The experimental results demonstrate that our proposed Glance-Focus model achieves state-of-the-art performance on all four datasets. We further verify the effectiveness of our approach via model analysis, showing the outstanding performance of key components in our method. Moreover, both quantitative and qualitative results affirm that our proposed memory prompts play a significant *anchor* role in multi-event reasoning by bridging the gap between questions and videos.

## 2 Related Works

### 2.1 Video Question Answering

Early attempts[11, 49, 6, 51, 41] have made progress in improving complex spatial-temporal con-textual information modeling for VideoQA tasks. They usually extract the appearance and motion features of video units (such as clips or frames) separately and then let them interact with the question to find an answer. To capture question-related spatial and temporal clues, co-memory and co-attention mechanisms are proposed by MACN[11] and HME[6] respectively. B2A[35] uses questions as a bridge to connect the appearance and motion features to find the answers. These holistic models are effective at aggregating spatial and temporal context, making them well-suited for general one-step video reasoning tasks. However, they tend to encounter challenges when confronted with more intricate compositional questions[14].

Therefore many studies[23, 15, 30, 5, 44, 45] have delved into the exploration of hierarchical modeling strategies to address the challenges of multi-grained video content representation. HCRN[23] proposes using reusable units stacked hierarchically to gradually extract frame-level to video-level features conditioned by motion and question. Graph-based methods such as L-GCN[15], HAIR[30], and VGT[45] utilize extra object detection models, like Faster R-CNN[37], to explicitly incorporate object-level features into the model. They further organize the objects into a unified graph for fine-grained reasoning. Although these hierarchical structures contain multi-grained clues, few of them explicitly model event-level structure of videos. This limitation poses a challenge in effectively handling high-level tasks, including event prediction, explanation, and counterfactual reasoning.

### 2.2 Video-Language Pre-trained Models

Recently, Transformers-based large video-language models (VLMs) [52, 25, 2, 8, 47, 50, 12, 1, 3, 40] have accomplished comparable or superior results on downstream tasks like VideoQA. These models harness the power of large-scale image or video-text pairs collected from the Internet to pre-train cross-modal correspondence models. They commonly employ proxy tasks like masked language modeling and masked frame modeling [25, 50, 40] or contrastive learning [47, 2] during pre-training. Some models like ActBERT[52] and BridgeFormer[12] focus on multi-grained semantics grounding by explicitly learning the alignment of objects and actions across modalities. However, applying these VLMs directly to Multi-Event VideoQA tasks is non-trivial. These models often necessitate sparse sampling of lengthy videos, leading to a notable loss of video semantics, and thus suffer from weak correspondence between questions involving high-level event concepts and intricate video content[9, 16]. To tackle this challenge, some retrieval-based cross-modal methods [48, 19, 36] have been proposed recently to efficiently extract the keyframes of the video for reasoning. Nonetheless, the use of fixed keyframes may pose difficulties when dealing with multi-grained questions. Instead, we propose to employ the dynamic and adaptable event memories for representing lengthy videos.

### 2.3 Multi-Event Modeling

Multi-event modeling is an essential ability for household robots to assist humans in completing complex tasks. However, explicit multi-event modeling still needs to be explored for VideoQA task. TSEA[9] proposes a heuristic method to extract event-level features of videos. Several studies targeting at multi-activity recognition and detection task[15, 34, 33] employ graph structures to model complex relationships among multiple activities. P-GCN[15] extracts activity proposals from videos and utilizes graph models to integrate context. Ego-Topo[33] organizes egocentric videos into several

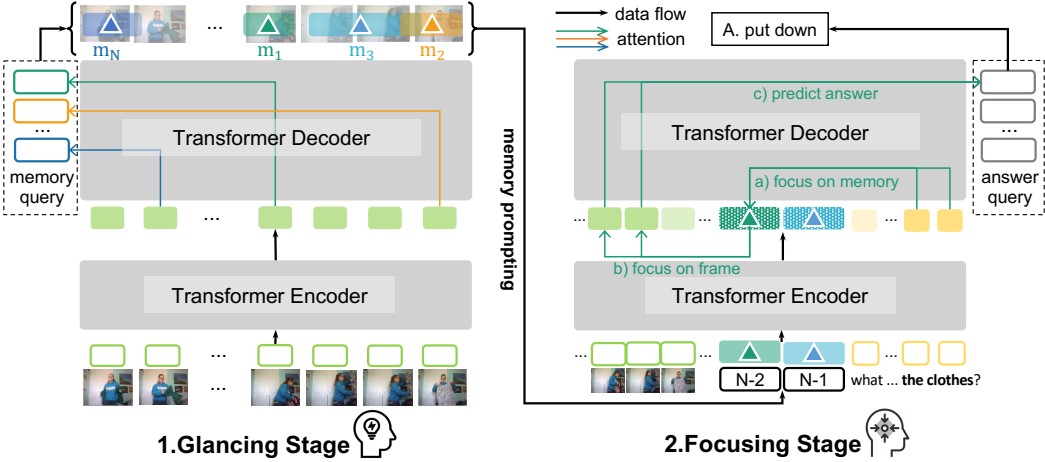

Figure 2: The complex multi-event reasoning task is structured into two stages: Glancing and Focusing. At the Glancing stage, a set of unordered event memories are obtained by aggregating relevant input video frames. At the Focusing stage, these event memories are reorganized according to their sequential order. The resulting memory prompts act as bridges, allowing model progressively focus on the related video frames for answer prediction using multi-level cross-attention mechanism.

zones and learns the links between these zones for scene affordances and future action prediction. Bridge-prompt[27] leverages the continuity of the natural language to concatenate multiple discrete events by constructing a series of text prompts. Similar to the multi-activity detection task, our goal is to obtain a set of high-level key event from the video. However, directly training an action detection model requires additional event-level annotations, which may not always be available. Furthermore, it is impractical to use a closed set of actions to encompass all possible events in the real-world. By contrast, our proposed approach learns dynamic event memories from videos through either unsupervised generation or supervised set-matching methods, depending on the availability of event labels during the training process.

## 3 Approach: The Glance-Focus Model

We first give an overview of our model in Section 3.1. Next, we describe in detail the two stages of our model, the Glancing stage and the Focusing stage in Section 3.2 and Section 3.3 respectively. Finally, in Section 3.4 we illustrate the overall loss functions.

### 3.1 Overview

Given a video clip $V$ and a question $q$, a VideoQA model predicts the answer $y$, formulated as follows:

$$\hat{y} = \arg\max_{y \in \mathcal{A}} \mathcal{F}_\theta(y|q, V, \mathcal{A}), \tag{1}$$

where $y$ is the predicted answer chosen from the candidate answers denoted as $\mathcal{A}$, i.e., answer vocabulary for open-ended questions or choices for multi-choice questions. $\theta$ is the trainable parameters of a VideoQA model $\mathcal{F}$.

We propose a novel Glance-Focus VideoQA model with the Transformer Encoder-Decoder framework[39]. As shown in Figure 2, the model answers the question with two stages: 1) At the Glancing stage, a set of episode memories are queried from the input video; 2) At the focusing stage, the model reorganizes the memories into memory prompts and they will serve as anchors to localize the question-related video content by multi-level focusing.

**Input representations.** For the input multi-event video of length $T$, we use a pretrained visual backbone to extract its features $\mathbf{X} \in \mathbb{R}^{T \times D}$ with dimension $D$. The video features are fed into the model both in glancing and focusing stages. For questions, following [31], we tokenize them using Byte-Pair Encoding (BPE) tokenization, and embed them with pretrained RoBERTa language

model[31] to get textual tokens $\mathbf{w} = \{w_1, ..., w_L\}$ with question length $L$. Then they are projected into the same dimension feature space as video features with a projector $\mathcal{G}(.)$ to get question embeddings $\mathbf{L} = \mathcal{G}(\mathbf{w}) \in \mathbb{R}^{L \times D}$. The question embeddings are fed into the model only at focusing stage.

## 3.2 Glancing Stage

It is challenging for the VideoQA model to directly find answers from lengthy videos containing multiple complex events. To efficiently establish the corresponding relationships between the high-level event concepts of the questions and the low-level video unit features, we propose employing a quick glance strategy that effectively extracts high-level episode memories from the video to guide the subsequent reasoning.

To obtain the episode memory, a direct approach would be training an action detection model to extract a set of actions from the video. However, this may not be applicable in certain cases where video data lacks event annotations. Moreover, it is impractical to encompass all possible events in the real world with a close-set vocabulary. To address these issues, we propose a more flexible and practical solution which can automatically learn a set of dynamic memories $\mathbf{M} = \{\mathbf{m_1}, ..., \mathbf{m_N}\} \in \mathbb{R}^{N \times D}$ to summarize the key $N$ events occurring in the videos. Given a group of randomly initialized memory queries, we train a Transformer encoder-decoder[39] to update them by gradually aggregating the related encoded video features $\mathbf{X}$ through cross-attention interactions. Specifically, we design both unsupervised generation and supervised matching approaches depending on whether event-level annotations are available.

**Unsupervised event memory generation.** In this case, we employ the idea of information maximization[21, 29] from the unsupervised learning to generate a set of event memories. Our goal is to generate a collection of event memories. To ensure that this set comprehensively captures all the crucial events within the video, we impose two vital constraints. 1) **Individual Certainty**. Each individual event is encouraged to be distinctive, thereby discouraging individual frames from forming separate and independent events. 2) **Global Diversity**. We aim to maintain diversity within the event set, preventing all video frames from being clustered into only one single event.

Assuming that there are $C$ types of events that will occur in the video. For individual certainty, we suppose higher classification confidence score. Namely, the ideal category distribution is similar to one-hot encoding[21]. Specifically, a classifer header $f_c$ of a linear layer with softmax is used to predict the event classification logits $\hat{\mathbf{p}}_m = f_c(\mathbf{m})$. The individual certainty loss is denoted as

$$\mathcal{L}_{cert}(f_c; \mathbf{M}) = -\mathbb{E}_{\mathbf{m} \in \mathbf{M}} \sum_{i=1}^{C} \hat{\mathbf{p}}_m^i \log \hat{\mathbf{p}}_m^i. \tag{2}$$

To ensure global diversity, we consider both the diversity of semantics and temporal spans within the event set. To promote semantic diversity, we assign category labels evenly across all memories. Namely, in contrast to the above one-hot classification logits of each event, we aim for an even distribution. The following $\mathcal{L}_{cls}$ is minimized to constitute the semantic diversity loss

$$\mathcal{L}_{cls}(f_c; \mathbf{M}) = \sum_{i=1}^{C} \hat{\mathbf{p}}^i \log \hat{\mathbf{p}}^i, \tag{3}$$

where $\hat{\mathbf{p}} = \mathbb{E}_{\mathbf{m} \in \mathbf{M}}[f_c(\mathbf{m})]$ is the mean output embedding of the all event memories of a video sample. Next, to ensure temporal span diversity, we pursue a reduction in the temporal overlap of all events. We use a temporal header $f_t$ of 3-layer FFN with ReLU to predict the normalized temporal center coordinate $\hat{\delta}$ and width $\hat{w}$ of events, $\hat{\tau} = f_t(\mathbf{m}) = [\hat{\delta}, \hat{w}] \in [0, 1]^2$. The temporal IoU loss formulated as,

$$\mathcal{L}_{iou}(f_t; \mathbf{M}) = \mathbb{E}_{\mathbf{m} \in \mathbf{M}} \sum_{i,j=1}^{N} IoU(\hat{\tau}_i, \hat{\tau}_j), \tag{4}$$

where IoU(.) is the normalized temporal IoU[24] given the paired temporal prediction of events. The overall global diversity loss is the weighted sum of $\mathcal{L}_{cls}$ and $\mathcal{L}_{iou}$

$$\mathcal{L}_{div}(f_c; f_t; \mathbf{M}) = \lambda_{cls}\mathcal{L}_{cls} + \lambda_{iou}\mathcal{L}_{iou}. \tag{5}$$

**Supervised event memory matching.** For datasets with ground-truth event-level labels, we similarly use the set prediction via bipartite matching [4, 24] to extract event memories. Since the output of the Transformer is unordered, we do not have a one-to-one correspondence between the predicted event memories and the ground-truth events. Following [4, 24], we use the Hungarian algorithm to find the optimal bipartite matching results. Given the padding ground-truth event labels $E = \{e_i\}_{i=1}^N$, $e_i = \{c_i, \tau_i = [\delta_i, w_i]\}$, where $c$ and $\tau$ are class labels and the temporal labels respectively, the matching cost $\mathcal{C}_{match}$ is defined as:

$$\mathcal{C}_{match} = \sum_{i=1}^{N}[-\mathbf{1}_{c_i \neq \emptyset}\lambda_{cls}\hat{p}_{\pi(i)}^{c_i} + \mathbf{1}_{c_i \neq \emptyset}\lambda_{L1}||\tau_i - \hat{\tau}_{\pi(i)}||], \tag{6}$$

where $\pi \in \Pi_N$ is the permutation of $N$ event memories. Then the goal is to find the optimal $\pi$ to minimize the matching cost.

**Memory prompting.** At the glancing stage, the output is a set of unordered event memories $\mathbf{M} = \{\mathbf{m_1}, ...\mathbf{m_N}\}$. We reorganize them into $\widetilde{\mathbf{M}}$ in the sequential order according to their temporal center coordinate $\{\delta_i\}$. The temporal positional embedding $\mathbf{P}_t \in \{\Phi(i)|i \in [1, N]\}$ is added to the corresponding events to indicate their temporal positions in the video. And they will serve as the memory prompts during multi-level attention at the following focusing stage.

### 3.3 Focusing Stage

It is difficult to directly associate the high-level event semantics in the questions with corresponding low-level videos. Thus the event memories extracted through video glancing serve as useful anchors to prompt the model quickly locate event segments related to the questions. Video features $\mathbf{X}$, memory prompts $\widetilde{\mathbf{M}} + \mathbf{P}_t$, and question embeddings $\mathbf{L}$ are concatenated as input. The Transformer Encoder embeds these embeddings from different sources into a joint space $[\mathbf{X}_{enc}, \mathbf{M}_{enc}, \mathbf{L}_{enc}]$.

**Multi-level cross attentions.** Given a set of answer queries $\mathbf{a} \in \mathbb{R}^{N \times D}$, the Transformer Decoder exploits multi-level cross attention to replace the original multi-head cross-attention to guide the model gradually focus on the question-related visual clues in the video.

a) *Focus on memory.* Given questions with high-level event semantics, we first use question embeddings as queries to interact with memory prompts to guide the model focus on the corresponding event memories $\mathbf{M}_{foc}$:

$$\mathbf{Q} = \mathbf{L}_{enc}, \mathbf{K} = \mathbf{V} = \mathbf{M}_{enc} \tag{7}$$
$$\mathbf{M}_{foc} = Multihead(\mathbf{Q}, \mathbf{K}, \mathbf{V}). \tag{8}$$

b) *Focus on frame.* Then the selected event memories prompt the model to localize the corresponding video frames $\mathbf{X}_{foc}$ as visual clues for specific reasoning:

$$\mathbf{Q} = \mathbf{M}_{foc}, \mathbf{K} = \mathbf{V} = \mathbf{X}_{enc} \tag{9}$$
$$\mathbf{X}_{foc} = Multihead(\mathbf{Q}, \mathbf{K}, \mathbf{V}). \tag{10}$$

c) *Predict the answer.* Finally, the answer queries find the answers $\hat{\mathbf{a}}$ through reasoning across the focused video frames:

$$\mathbf{Q} = \mathbf{a}, \mathbf{K} = \mathbf{V} = \mathbf{X}_{foc} \tag{11}$$
$$\hat{\mathbf{a}} = Multihead(\mathbf{Q}, \mathbf{K}, \mathbf{V}). \tag{12}$$

Similarly to classifier header $f_c$, an answer classifier header $f_a$ of a linear layer with softmax is used to predict the answer label from the last token of answer queries $\hat{y} = f_a(\hat{\mathbf{a}})$.

### 3.4 Loss Functions

The first part of loss functions is the answer classification cross-entropy loss $\mathcal{L}_{qa}$. The overall unsupervised loss is defined as a linear combination:

$$\mathcal{L}_{uns}(\mathcal{F}_\theta; q; V; \mathcal{A}; y) = \mathcal{L}_{qa} + \lambda_{cert}\mathcal{L}_{cert} + \mathcal{L}_{div}. \tag{13}$$

Under the situation with ground-truth event-level labels, we turn to use them for supervision by minimizing the event classification cross-entropy loss $\mathcal{L}_{cls}$ and the temporal span $L1$ loss $\mathcal{L}_{L1}$ between the matching memories with the ground-truth events:

$$\mathcal{L}_{sup}(\mathcal{F}_\theta; q; V; \mathcal{A}; y; E) = \mathcal{L}_{qa} + \lambda_{cls}\mathcal{L}_{cls} + \lambda_{L1}\mathcal{L}_{L1}. \tag{14}$$

## 4  Evaluation

### 4.1  Datasets

We evaluate our model on various recently proposed Multi-Event VideoQA datasets. The comparison of them is shown in Table 1. Note that we use the v2 version of the AGQA dataset, which has more balanced distributions, as the dataset creator recommended. We choose these benchmarks for a broad coverage of: **Video content**. Including household and general activities, *etc*. **Video duration**. The average video length ranges from 12 seconds to 44 seconds. **Event complexity**. All these videos contain multiple events or actions, with the average event number ranging from 2.7 to 8.8. **Event complexity**. The number of event categories varies from 50 to 793.

Table 1: Comparison of the Multi-Event VideoQA benchmarks.

| Datasets | # Avg. Events/Video | Video Length | #Event Classes | #Videos | # QAs |
|---|---|---|---|---|---|
| STAR[42] | 2.7 | 12s | 157 | 22K | 60K |
| AGQA v2[14] | 6.8 | 30s | 157 | 9.7K | 2.27M |
| EgoTaskQA[16] | 5 | 25s | 793 | 2K | 40K |
| NExT-QA[43] | 8.8 | 44s | 50 | 5.4K | 52K |

### 4.2  Implementation Details

For each benchmark, we follow standard protocols outlined by prior works for data processing, metrics, and settings to ensure fair comparisons. For video feature extraction, we use frozen S3D model [46] pre-trained on HowTo100M[32] on STAR dataset and C3D[38] feature on EgoTaskQA dataset following[47, 16]. We also try Faster-RCNN model[37] pretrained on the Visual Genome[22] and CLIP(ViT-B/32) for more challenging AGQA and NExT-QA datasets. We employ a standard 2-layer, 8-head Transformer Encoder-Decoder with hidden size $D$ of 512 as the backbone for our Glance-Focus model. According to the statistics of common event categories, the query number $N$ is set to 10 since the maximum average number of events per video in the current dataset is approximately 8.8. And the event class number $C$ is set to the ground truth overall class number in the dataset or 100 if it is unknown by default. For training details, we use dropout of 0.1, and initialize model weights using Xavier init[13]. Adam optimizer[20] is used with a learning rate of 5e-6 to optimize model parameters. All experiments are conducted on an NVIDIA GeForce RTX 3090Ti GPU. More details can be found in the supplementary material.

### 4.3  Comparison with State-of-the-arts

We conduct a comprehensive comparison with current state-of-the-arts (SOTA) VideoQA methods, including traditional spatial-temporal models and large vision language models. We compare our Glance-Focus model (shorted as GF model in the following) with them on all four datasets. The results are shown in Table 2, 3, 4 and 5 respectively. Overall, our proposed GF model achieves SOTA results on all benchmarks, and obtains significant advantages in both the STAR and the EgoTaskQA dataset. We will analyze different datasets one by one as follows. Note that in Table 2-5 the red numbers highlight the significant gains compared to prior SOTA results.

For **STAR** dataset, our proposed model obtains excellent reasoning performance shown in Table 2. When facing all four different types of questions, GF significantly surpasses the previous SOTAs, especially on the Sequence category. Such results reflect that our model has strong temporal contextual modeling ability. What's more, note that our model trained without ground-truth event labels, *i.e.*, GF(uns), has achieved comparable results with the one trained under supervision, *i.e.*, GF(sup). It indicates that our proposed unsupervised memory generation in the Glance stage can effectively extract key events with strong discriminability.

Table 2: QA accuracies of SOTA methods on STAR test set (* indicates the val set results).

| Model | Interaction | Sequence | Prediction | Feasibility | Mean |
|---|---|---|---|---|---|
| ClipBERT[25] | 39.81 | 43.59 | 32.34 | 31.42 | 36.70 |
| RESERVE-B[50] | 44.80 | 42.40 | 38.80 | 36.20 | 40.50 |
| Flamingo-9B[1] | - | - | - | - | 43.40 |
| AIO[40]* | 47.53 | 50.81 | 47.75 | 44.08 | 47.54 |
| Temp[ATP][3]* | 50.63 | 52.87 | 49.36 | 40.61 | 48.37 |
| MIST[10]* | 55.59 | 54.23 | 54.24 | 44.48 | 51.13 |
| GF(uns) | 51.91 | **63.06**(+8.83) | **54.89** | 45.57 | 53.86(+2.73) |
| GF(sup) | **56.10** | 61.27(+7.04) | 52.65 | **45.74** | **53.94**(+2.81) |

We also evaluate our model in the normal split of the **EgoTaskQA** dataset. From Table 3, we can find that our model outperforms SOTA methods on almost all question types. Our unsupervised GF model has strong performance especially on the intent and explanatory question types. These questions evaluate the understanding of the humans daily activities, which are all necessary for building the household robots. And the model trained under supervision achieves strong performance especially on object and predictive question types, which indicates that by explicitly modeling the events happening in the video, the model can obtain stronger event semantic understanding ability, namely, to recognize the interacted objects and to predict the next action.

Table 3: QA accuracies of SOTA methods on EgoTaskQA normal split.

| | Category | VisualBERT[26] | ClipBERT[25] | HCRN[23] | GF(uns) | GF(sup) |
|---|---|---|---|---|---|---|
| Scope | world | 39.73 | 42.15 | 44.27 | 44.65 | **46.53** |
| | intent | 44.51 | 40.94 | 49.77 | **52.07**(+2.30) | 51.88 |
| | multi-agent | 26.29 | 27.63 | 31.36 | 33.17 | **34.31** |
| Type | descriptive | 41.99 | 38.45 | 43.48 | 44.48 | **46.68**(+3.20) |
| | predictive | 30.37 | 31.50 | 36.56 | 39.00 | **40.36**(+3.80) |
| | counterfactual | 41.99 | 46.75 | **48.00** | 46.66 | 46.80 |
| | explanatory | 37.42 | 42.39 | 40.60 | **42.45** | 42.18 |
| Semantic | action | 15.02 | 22.91 | 14.92 | 15.68 | **15.68** |
| | object | 23.26 | 21.80 | 45.31 | 45.97 | **50.98**(+5.67) |
| | state | 59.20 | 54.36 | 68.28 | 70.44 | **71.23** |
| | change | 68.27 | 66.58 | 67.38 | 68.69 | **69.71** |
| Overall | open | 24.62 | 27.70 | 30.23 | 31.26 | **32.79**(+2.56) |
| | binary | 68.08 | 67.52 | 69.42 | 69.80 | **70.25** |
| | all | 37.93 | 39.87 | 42.20 | 43.06 | **44.27**(+2.07) |

As for **AGQA** dataset, from Table 4, our model also achieves the SOTA result with object-level video features, *i.e.*, GF(sup) with Faster R-CNN[37] backbone. And the one trained with S3D features also gains comparable results. We show several representative reasoning types of the dataset. Our model obtains better results on some challenging question types, such as obj-rel composition questions and temporal superlative questions like "*the longest...*". Overall, our model shows more clear advantage over SOTA on the challenging Open questions, compared to the relatively simple Binary questions for which the model could gain a 50% accuracy rate by blind guessing. The complete breakdown results on all kinds of question types and a more thorough analysis can be found in the Supplementary Material.

Table 4: QA accuracies of SOTA methods on AGQA v2 test set.

| Method | HME[6] | HCRN[23] | AIO[40] | Temp[3] | MIST[10] | GF(uns) | GF(uns) | GF(sup) |
|---|---|---|---|---|---|---|---|---|
| Backbone | C3D | C3D | ViT | CLIP | CLIP | S3D | FasterRCNN | FasterRCNN |
| obj-rel | 37.42 | 40.33 | 48.34 | 50.15 | 51.68 | 52.93 | 54.31 | **54.96**(+3.28) |
| superlative | 33.21 | 33.55 | 37.53 | 39.78 | 42.05 | 42.78 | 42.90 | **44.62**(+2.57) |
| sequencing | 49.77 | 49.70 | 49.61 | 48.25 | **67.24** | 53.03 | 51.97 | 53.24 |
| exist | 49.96 | 50.01 | 50.81 | 51.79 | **60.33** | 58.31 | 58.08 | 59.13 |
| duration | 47.03 | 43.84 | 45.36 | 49.59 | **54.62** | 50.86 | 52.02 | 52.80 |
| act. recog. | 5.43 | 5.52 | 18.97 | 18.96 | 19.69 | **22.08**(+2.39) | 16.38 | 14.17 |
| open | 31.01 | 36.34 | - | - | 50.56 | 53.06 | 55.66 | **56.07**(+5.51) |
| binary | 48.91 | 47.97 | - | - | **58.28** | 53.61 | 53.52 | 54.17 |
| all | 39.89 | 42.11 | 48.59 | 49.79 | 54.39 | 53.33 | 54.59 | **55.08** |

We also evaluate our model on the challenging **NExT-QA** dataset based on different video backbones. Note that the NExT-QA dataset does not provide event annotations. Therefore, we exploit the proposed unsupervised memory generation in the Glance stage. As shown in Table 5, without event-level supervision or applying any extra action detection model, the unsupervised GF models with the same backbone as other methods still achieve SOTA performances. Such results further justify the strong multi-event reasoning ability of our method, considering that NExT-QA is currently the most complex Multi-Event VideoQA benchmark with longer video duration (44s) and richer events (8.8 actions per video).

What's more, we also report the breakdown results by the different question types. According to the results, GF has strong event reasoning ability especially on Descriptive (Acc@D) questions, improving the accuracy by 3.61%, and also on event relationship reasoning, achieving 2.31% accuracy gain on Causal (Acc@C) questions. As recommended in ATP[3], we additionally report results on its proposed ATP-hard subset of the NExT-QA validation set. This subset filters out those "easy" questions that can be answered with a single frame. The results in Table 5 show that our GF models significantly surpass HGA[17] models on this subset by improving 5.22% accuracy overall. The above results indicate that compared to HGA and other methods that directly model the correspondence between question and long video through co-attention, our method using event memory as an intermediate bridge to establish such correlation for reasoning is more efficient.

Table 5: Accuracy (%) comparison on NExT-QA validation set and the ATP-hard subset[3]. Acc@C, T, D, denote accuracy for Causal, Temporal, and Descriptive questions respectively.

| Methods | Backbone | val | | | | val (ATP-hard subset) | | |
|---|---|---|---|---|---|---|---|---|
| | | Acc@C | Acc@T | Acc@D | Acc@All | Acc@C | Acc@T | Acc@All |
| ATP[3] | CLIP | 48.30 | 49.30 | 65.00 | 49.20 | 19.60 | 22.60 | 20.20 |
| VQA-T[47] | S3D | 49.60 | 51.49 | 63.19 | 52.32 | - | - | - |
| IGV*[28] | C3D | 48.56 | 51.67 | 59.64 | 51.34 | - | - | - |
| Temp[ATP][3] | CLIP | 53.10 | 50.20 | 66.80 | 54.30 | 38.40 | 36.50 | 38.80 |
| HGA[17] | C3D | 46.26 | 50.74 | 59.33 | 49.74 | 43.30 | 45.30 | 44.10 |
| MIST[10] | CLIP | 54.62 | 56.64 | 66.92 | 57.18 | - | - | - |
| GF(uns) | S3D | 51.78 | 54.71 | 63.96 | 54.62(+2.30) | 48.28 | 49.95 | 48.96 |
| GF(uns) | C3D | 53.59 | 55.71 | 66.80 | 56.33(+4.99) | 48.50 | 49.52 | 48.92 |
| GF(uns) | CLIP | **56.93**(+2.31) | **57.07** | **70.53**(+3.61) | **58.83** | **48.65**(+5.35) | **50.27**(+4.97) | **49.32**(+5.22) |

## 4.4 Model Analysis

In this section, we evaluate the effectiveness of our proposed modules by comparing different variants of the GF models and various baselines. We first introduce the baselines and GF variants as follows. **Baselines**: 1) **MDETR**[18] which utilizes same 2-layer Transformer Encoder-Decoder architecture. 2) An **Oracle** model with ground-truth event labels concatenated to the questions as textual input as an upper-bound model. **GF Variants**: 1) **Glance-only**. Model which uses standard cross-attention to replace the multi-level attention strategy. 2) **Detect-Focus**. Apply a frozen action detection model, such as SlowFast[7], to replace the Glance stage of the model. We try to keep the model parameters consistent.

**Effect of memory prompting.** Firstly, the performance of the Oracle model significantly improves compared to the baseline model. It convinced us that event memories do play an important prompting role in the Multi-Event VideoQA task. Besides, it can be found that with event memory generation at the glancing stage, the Glance-only model outperforms the MDETR baseline and the model using a frozen action detection (Detect-Focus) with a large gap on both datasets. Such results reflect the effectiveness of our proposed memory-prompting strategy. Compared to additionally fine-tuning an action detection model, our proposed Glance strategy can adaptively generate key event/action features of the long videos without any annotations.

What's more, we evaluate the effectiveness of different loss functions that are designed during the event memory extraction in Table 7. We carefully evaluate all pairs of losses on STAR, EgoTaskQA and AGQA datasets. Firstly, the model with complete losses achieves the best results on both unsupervised and supervised settings of all datasets. It indicates that each loss plays its role in generating the event memories. Secondly, under the unsupervised setting, the $\mathcal{L}_{iou}$ shows an

important role in controlling the temporal prediction of the events. For supervised settings, the $\mathcal{L}_{cls}$ shows a more important role for the event semantic supervision helps cross-modal grounding. Please see more discussions on the loss functions in the Supplementary Material.

**Effect of multi-level attention.** From Table 6, compared to the Glance-only model using standard cross-attention, the Glance-Focus model with our designed multi-level attention shows stronger reasoning ability on both STAR and EgoTaskQA datasets. Taking an example from STAR dataset, we further visualize the attention weights in different levels in Figure 3, the darker the color, the greater the weight. The attention map in the first row shows the correspondence of the question embeddings with the generated memories. In this level, the model focuses on the most related event "*Throwing a towel*". The attention map in the second row is the correspondence between the focused event and the video embeddings. At this level, the model correctly attends to the most related keyframes for answer prediction.

Table 6: Module ablations on STAR and EgoTaskQA datasets.

| Model | STAR | EgoTaskQA |
|---|---|---|
| MDETR | 46.26 | 39.68 |
| Detect-Focus | 49.85 | 42.62 |
| Glance-only | 53.12 | 42.70 |
| Glance-Focus | **53.94** | **44.27** |
| Oracle(Upper) | 55.19 | 49.52 |

Table 7: Ablation study of the loss functions on all datasets.

| Uns/Sup | $\mathcal{L}_{cls}^{uns}$ | $\mathcal{L}_{iou}$ | $\mathcal{L}_{cert}$ | $\mathcal{L}_{cls}^{sup}$ | $\mathcal{L}_{L1}$ | AGQA | STAR | EgoTaskQA |
|---|---|---|---|---|---|---|---|---|
| Uns | ✓ | ✓ | | | | 54.12 | 53.62 | 42.69 |
| Uns | ✓ | | ✓ | | | 52.89 | 51.35 | 41.77 |
| Uns | | ✓ | ✓ | | | 54.27 | 53.72 | 42.85 |
| Uns | ✓ | ✓ | ✓ | | | **54.59** | **53.86** | **43.06** |
| Sup | | | | ✓ | | 54.26 | 53.69 | 43.87 |
| Sup | | | | | ✓ | 53.19 | 53.03 | 42.64 |
| Sup | | | | ✓ | ✓ | **55.08** | **53.94** | **44.27** |

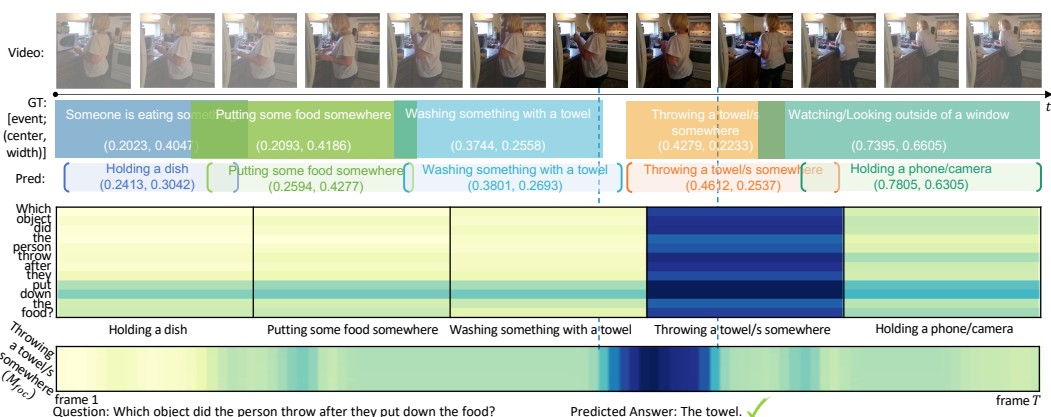

Figure 3: Visualization of the event memory generation and the multi-level attention map.

## 5 Conclusion

Our proposed Glance-Focus model provides an innovative approach to the challenging Multi-Event VideoQA task. With a set of event memories as prompts, the model can efficiently establish the correspondence between high-level questions and low-level video frames. Our experiments has shown that event memories play a crucial role in locating question-related visual clues for reasoning. The proposed Glance-Focus strategy can obtain a complete and compact representation of videos without additional annotations, which can be extended to more video comprehension tasks, such as video captioning/summarization. We hope this study can inspire more future research in the field of Multi-Event Video Understanding.

## Acknowledgements

This work is partially supported by National Key R&D Program of China No. 2021ZD0111901, and Natural Science Foundation of China under contracts Nos. U21B2025, U19B2036.

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
