# Glance and Focus: Memory Prompting for Multi-Event Video Question Answering Supplementary Material

**Ziyi Bai, Ruiping Wang, Xilin Chen**
*ziyi.bai@vipl.ict.ac.cn*, *{wangruiping, xlchen}@ict.ac.cn*

In this supplementary document, we introduce more details of evaluation settings (Section 1), more experimental results (Section 2), and more visualizing cases (Section 3). We also discuss the limitations of our method and future works in more details (Section 4).

## 1  More Details about Evaluation Setting

### 1.1  Data Processing

As mentioned in Section 4.2[1] of our main paper, we try various popular video backbones following the current VideoQA works.

- Frozen **S3D**[18] model pre-trained on HowTo100M[10] following [19].
- **C3D**, which is commonly used for many spatial-temporal VideoQA models, such as HCRN[8] and HGA[6]. We only take the frame-wise appearance feature, which is the *pool5* output of ResNet[5] feature as input.
- **Faster R-CNN**[13] pre-trained on Visual Genome[7], which is usually used for fine-grained video content reasoning[17, 16]. Specifically, features of 10 detected objects with the highest confidence scores of each sampled frame are concatenated as the whole visual features.
- **CLIP** (ViT-B/32) [12], commonly used cross-modal pre-trained models, which is proved to have strong generalization on VideoQA task in ATP[2] and MIST[4].

Our model can easily adapt to various video backbones. From the comparison results in Table 4 and 5, generally, the local video features like Faster R-CNN and CLIP achieve better results than the global video features like S3D and C3D features. However, our Glance-Focus (GF) model with different visual backbones achieves SOTA performances compared to the methods using the same backbones.

### 1.2  Evaluation Metrics

We evaluate our approach on four datasets in the main paper and an additional ActivityNet-QA dataset in this Supplementary Material. We use QA accuracy as the metric for evaluation. Specifically, we assign a score of 1 if the output answer matches the corresponding ground truth, and a score of 0 otherwise. When evaluating different question types within each dataset, we calculate the accuracy rate separately for each type of question.

### 1.3  Implementation Details

**Supervised event memory matching.** As illustrated in Section 3.2, with event-level annotations, we explicitly extract a set of event memories using the bipartite matching between the prediction and the ground-truth events. Note that we only use these event-level annotations as supervision during

---

[1]For better understanding, we denote the references to sections, figures, and tables in the main paper in blue.

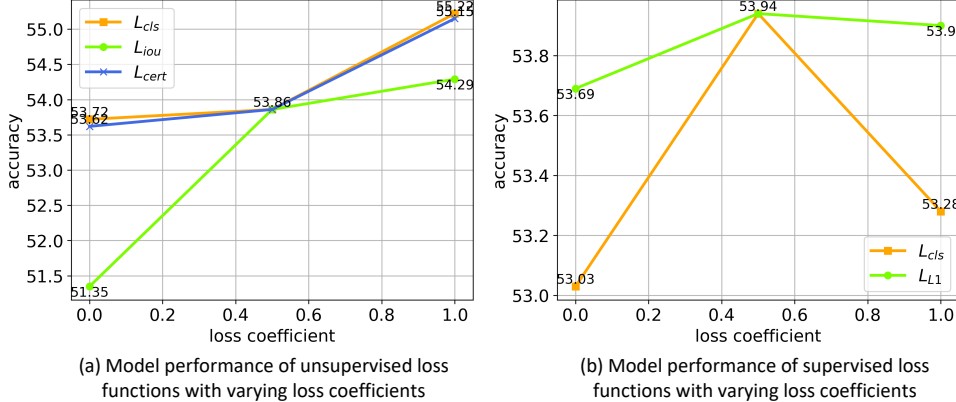

(a) Model performance of unsupervised loss functions with varying loss coefficients

(b) Model performance of supervised loss functions with varying loss coefficients

Figure 1: Ablation study on loss functions with varying loss coefficients. (a) $L_{cls}$, $L_{iou}$ and $L_{cert}$ in unsupervised loss. (b) $L_{cls}$ and $L_{L1}$ in supervised loss.

training. And the cost coefficients of semantic consistency $\lambda_{cls}$ and temporal consistency $\lambda_{L1}$ are set to 1 by default.

**Loss coefficients.** In the main paper, we design different loss functions in Section 3.2. The coefficient of each loss function is set to 0.5 by default. Namely, $\lambda_{cls}$, $\lambda_{iou}$, $\lambda_{cert}$ are set to 0.5 in unsupervised loss, and $\lambda_{cls}$, $\lambda_{L1}$ are set to 0.5 in supervised loss. We also conduct the ablation study on them in Section 2.

**Transformer Encoder-Decoder.** We employ a standard 2-layer 8-head Transformer Encoder-Decoder[15] as the backbone. Through experiments, it was found that the 6-layer model has better performance on the EgoTaskQA dataset.

## 2 More Experimental Results

### 2.1 Ablation Study on Different Losses

First, we analyze the effects of different loss functions on model performance. For each loss function, while keeping the other coefficients as default settings, we evaluate the model's performance using varying loss coefficients, specifically, $\lambda = [0.0, 0.5, 1.0]$. The evaluation is conducted with the STAR dataset. The results are shown in Figure 1. When the coefficient of any loss function is 0, the performance of the model decreases, which indicates their efficiency in event memory extraction. Among three unsupervised loss functions, the $L_{iou}$ that ensures the temporal diversity of the generated event memories is the most important. Without it, there is a significant decrease in model performance. Besides, the model seems to gain stronger performance when setting the coefficient as 1.0. While for supervised losses, the model is more robust to the coefficients. The best coefficient is 0.5.

### 2.2 Ablation Study on Event Class Number

As mentioned in Section 4.2, the event class number $C$ is a predetermined hyper-parameter in our model, denoting the number of event classes that may occur in the video. We further conduct an ablation study to evaluate the impact of different choices for $C$. We separately set $C = [1, 50, 100, GT]$. The mean accuracies of the test split on the STAR[16] dataset are reported. From Table 1, setting $C = 1$ will severely affect the performance of the model while setting $C$ to a random reasonable number will only slightly affect the performance of the model. Thus, the model is not sensitive to the number of categories and it can easily handle the scenarios when the dataset has no event-level annotations.

Table 1: Ablation study on the number of event categories.

| C | Acc |
|---|---|
| 1 | 51.68 |
| 50 | **54.58** |
| 100 | 53.15 |
| # GT classes(157) | 53.94 |

Table 2: QA accuracies of state-of-the-art (SOTA) methods on AGQA v2 test set. The proportion of questions is indicated in percentage. Faster R-CNN is abbreviated as FRC.

| | Question Types | PASC [9] | HME [3] | HCRN [8] | MIST [4] | GF(uns) -S3D | GF(uns) -FRC | GF(sup) -FRC |
|---|---|---|---|---|---|---|---|---|
| Reasoning | obj-rel(77.93%) | 37.84 | 37.42 | 40.33 | 51.68 | 52.93 | 54.31 | **54.96**(+3.28) |
| | rel-act(3.14%) | 49.85 | 49.90 | 49.86 | **67.18** | 52.24 | 51.75 | 52.09 |
| | obj-act(6.70%) | 50.00 | 49.97 | 49.85 | **68.99** | 53.94 | 53.13 | 54.44 |
| | superlative(15.40%) | 33.2 | 33.21 | 33.55 | 42.05 | 42.78 | 42.90 | **44.62**(+2.57) |
| | sequencing(13.14%) | 49.78 | 49.77 | 49.70 | **67.24** | 53.03 | 51.97 | 53.24 |
| | exist(14.55%) | 49.94 | 49.96 | 50.01 | **60.33** | 58.31 | 58.08 | 59.13 |
| | duration(1.98%) | 45.21 | 47.03 | 43.84 | **54.62** | 50.86 | 52.02 | 52.80 |
| | act. recog.(0.16%) | 4.14 | 5.43 | 5.52 | 19.69 | **22.08**(+2.39) | 16.38 | 14.17 |
| Semant. | object(80.14%) | 37.97 | 37.55 | 40.40 | 52.90 | 53.06 | 54.44 | **55.14**(+2.24) |
| | relationship(14.47%) | 49.95 | 49.99 | 49.96 | **60.76** | 55.76 | 55.36 | 56.25 |
| | action(5.39%) | 46.85 | 47.58 | 46.41 | **59.48** | 50.87 | 49.93 | 51.46 |
| Structure | query(50.37%) | 31.63 | 31.01 | 36.34 | 50.56 | 53.06 | 55.66 | **56.02**(+5.46) |
| | compare(15.02%) | 49.49 | 49.71 | 49.22 | **65.87** | 52.91 | 52.19 | 53.40 |
| | choose(13.60%) | 46.56 | 46.42 | 43.42 | 47.97 | **48.00** | 47.19 | 47.61 |
| | logic(5.10%) | 49.96 | 49.87 | 50.02 | **57.80** | 55.21 | 54.93 | 55.33 |
| | verify(15.91%) | 49.90 | 49.96 | 50.01 | 60.09 | 58.56 | 58.07 | **60.12** |
| Overall | open(50.37%) | 31.63 | 31.01 | 36.34 | 50.56 | 53.06 | 55.66 | **56.07**(+5.51) |
| | binary(49.63%) | 49.01 | 48.91 | 47.97 | **58.28** | 53.61 | 53.52 | 54.17 |
| | all(100.00%) | 40.18 | 39.89 | 42.11 | 54.39 | 53.31 | 54.59 | **55.08** |

## 2.3 Analysis of All Question Types on AGQA v2 Dataset

Here in Table 2, we present the complete comparison results on the AGQA v2 dataset between our Glance-Focus (GF) models with SOTA methods, including all reasoning types (Reasoning), querying semantics (Semantic.), question structure (Structure), and overall performance (Overall) as supplements of Table 4 in the main paper. Although MIST[4] outperforms other methods on certain question types like *rel-act, obj-act, etc*, the number of these types is relatively small in the test split, thus, the conclusion may not have generality. By contrast, both our GF models with the frame-level backbone (-frame) and object-level backbone (-obj) achieve better performances on types with a large number of questions, like *obj-rel (77.93%), object (80.14%), query (50.37%), open (50.37%), etc*. Therefore, our model gains better results overall. Specifically, GF achieves the most significant improvement on *open* question types *(+5.51)* compared to MIST. These types of questions require the model to answer the open-ended questions. It is more challenging than binary questions, which only require the model to do the binary choice between *yes* or *no etc*.

## 2.4 Evaluation of ActivityNet-QA Dataset

AcitivityNet-QA[21] is a long-term Video QA dataset. It contains 5.8K complicated web videos with an average duration of 180 seconds. Since the average event (action) of a video is merely about 1.4, which is outside the standard Multi-Event VideoQA benchmarks set, we only conduct a concise experimental analysis on it to evaluate whether the GF model can scale to longer videos. This dataset does not provide event annotations, therefore we apply unsupervised memory generation at the Glance stage. We exploit S3D and CLIP video backbones respectively. The results are shown in Table 3.

Our GF models achieve promising performances compared to SOTAs with the same backbone and training setting: *GF-S3D* vs. *VQA-T(w/o)* and *GF-CLIP* vs. *FrozenBiLM*. Note that the best model, FrozenBiLM[20], is pre-trained on the large dataset WebVid10M[1], which consists of 10 million video-text pairs and thus significantly benefits the final accuracy. However, our model shows comparable results without pre-training on such large-scale data, which indicates the generalization of GF to longer videos.

Table 3: Evaluation results on ActivityNet-QA.

| Method | Pre-training Data | Features | Acc |
|---|---|---|---|
| HGA[6] | - | C3D | 34.6 |
| LocAns[11] | - | C3D | 36.1 |
| VQA-T(w/o)[19] | - | S3D | 36.8 |
| VQA-T[19] | HowToVQA69M[19] | S3D | 38.9 |
| FrozenBiLM[20] | WebVid10M[1] | CLIP | **43.2** |
| GF-S3D(uns) | - | S3D | 37.0 |
| GF-CLIP(uns) | - | CLIP | 41.1 |

## 3 Extended Qualitative Experiments

To further validate the role of our generated event memories, we visualize the joint feature space of the word embeddings, frame embeddings, and memory prompts of an example in STAR[16] dataset. As shown in Figure 2, 10 memory prompts are distributed between word and frame features, which can help the model quickly locate the relevant video frames based on the specific question.

Besides, we visualize the predicted event memories, multi-level attention map, and the QA results predicted by our GF model in Figure 3. All the examples are from the STAR dataset [16] and the results are predicted by the supervised model.

We first compare the predicted events with the ground-truth events for each video example. The events are arranged in temporal order from top to bottom, represented by colored bars, and normalized within the range of $[0.0, 1.0]$, aligned with the sampled video frames. The results show a high overlap between the ground truth events and our predicted events, indicating relatively good correspondences among them. Although some event semantics are incorrect, such as "*Sitting in a chair*" instead of the ground-truth label "*Sitting on sofa/coach*" in the first example, their semantics are relatively similar and have little impact on subsequent reasoning tasks.

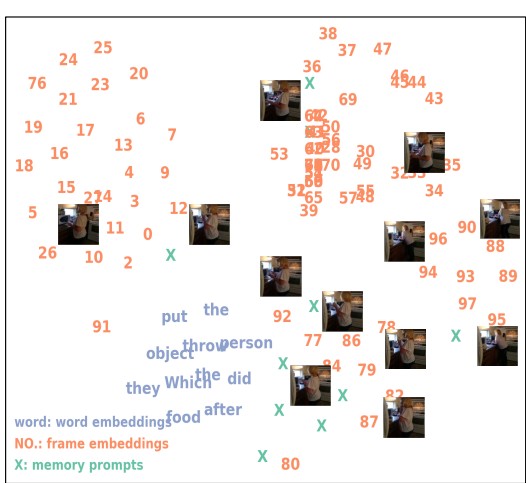

Figure 2: Visualization of the joint feature space.

Furthermore, we visualize the question-memory attention maps and memory-frame attention maps from the last attention layer of our model. The color of each attention position represents the weight assigned to it, with darker colors indicating higher weights. In the first example, the question guides the model to focus on the key event "*Taking a book from somewhere*". Subsequently, the focused memory leads the model to concentrate on the related video frames, enabling it to answer the question accurately. Similarly, in the second example, the question asks about the key event "*Putting a book somewhere*", and the focused event memory directs the model's attention towards the corresponding frames, leading to the correct answer of "*The book.*"

## 4 Limitations and Future Works

As outlined in Section 1, our approach uses a more flexible way to extract a set of event memories from videos. While under the supervised setting, the memories are explicitly required to have event semantics, in the unsupervised setting, there are no strong supervisions to ensure that this set of memories has event semantics. We use relatively straightforward constraints on generating the event memory. More intuitive ideas from the way human beings generate event memories are desired to be explored.

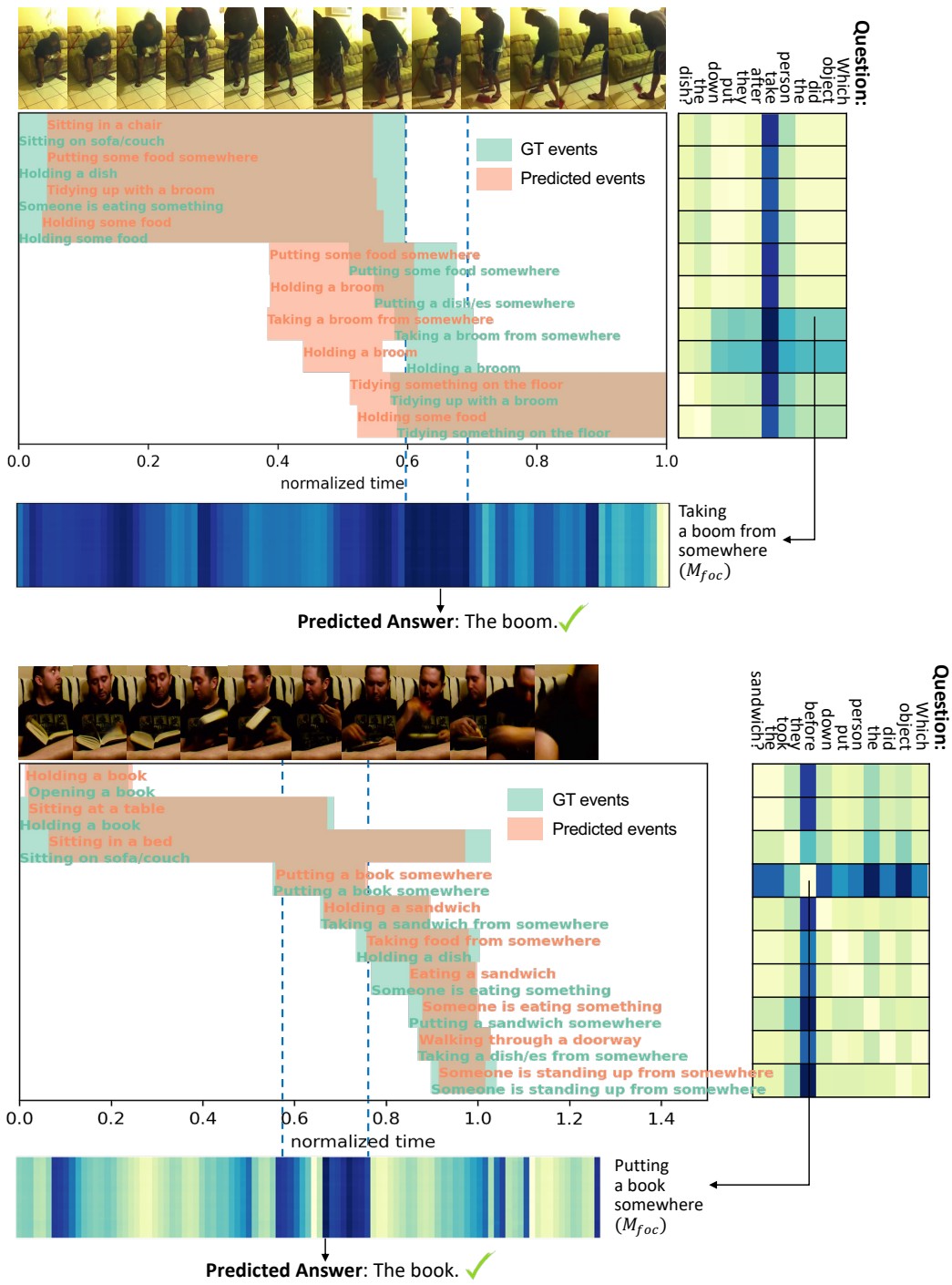

Figure 3: Comparison between the predicted events (orange bars) and the ground-truths (green bars). question-memory and memory-frame attention maps together with the QA results are visualized to show the reasoning process.

Besides, the current videos collected in the Video QA datasets are not long enough and the event is not complex enough to fully evaluate the model's multi-event reasoning ability. We look forward to further progress on multi-event reasoning research. Also, future works including keyframe selection strategies are considered to improve the model efficiency on long video understanding.

Our proposed memory prompt is related to prompt learning [14, 12], specifically the soft prompting approach proposed in [22]. We believe that our event memories serve as prompts to summarize the content of long videos, i.e., *"A video of event 1, event 2, ..., event N."*, which is implemented as the oracle model in our experiment. Compared with soft prompts that are generated without explicit constraints, our memory prompts are extracted through gradually aggregating related video frame embeddings by cross-attention and have stronger semantics due to our designed unsupervised loss. The experiments have demonstrated the importance of the memory prompts in localizing the question-related video content for event reasoning. In future work, we plan to explore further how memory prompts can be used in other video understanding tasks, like video captioning and summarization.