# OpenReview forum: "Glance and Focus: Memory Prompting for Multi-Event Video Question Answering"
_NeurIPS.cc/2023/Conference — NeurIPS 2023 poster_

### Official Review · Reviewer_yVqG · 2023-06-23

**Soundness:** 2 fair
**Presentation:** 2 fair
**Contribution:** 2 fair
**Rating:** 3
**Confidence:** 4

**Summary:**

This paper presents a model for video question answering called "Glance-Focus". The idea is to first "Glance" at a video to obtain a set of N dynamic memories, each represented as vectors of dimension D.

The memories can either be learned in an unsupervised way or a supervised way. In the unsupervised case, the paper uses a collection of losses encouraging the video frames be assigned to different memory cells. In the supervised case, where there are ground-truth event labels that are localized in time, the paper uses bipartite matching to match the dynamic memories with the ground truth event labels, and that forms the basis of the loss.

Next, the paper uses an encoder-decoder model. The encoder encodes the video features, memory cells, and question embeddings all together. Then, the decoder starts with a representation of the questio, and cross-attends to the memory cells, then to the video frames, and then to the answer.

The model is initialized from S3D pretrained on HowTo100M, and learned on STAR, AGQA, and EgoTaskQA. The model seems to do well on all the datasets.

--
update - I still agree with my concerns post-rebuttal and would like to keep my score.

**Strengths:**

* The core idea that this paper seems to be tackling is an important one. Videos are long and redundant, and dealing with this space better is an important area of research.
* The model seems to perform quite well on the STAR test set - (though, it should be noted, it was trained on the raw videos from that dataset, so it might be a bit of an unfair comparison versus the baselines).


**Weaknesses:**

To this reviewer, it is rather hard to evaluate this work as the model and the associated losses feel very complicated. I am pretty sure I don't understand how they all play together. I think the paper could do a much better job:
* Carefully ablating each one on a variety of datasets, extending Table 5 to consider all pairs of all losses.
* Carefully ablating the modules and architecture choices. Table 4 seems like a good start, but it should be noted that removing the "focus" module for instance would presumably change the parameter count or effective footprint of the model. As a result it's hard for this reviewer to draw any conclusions there.

It would also be great to explore a little bit the role of feature extraction. The paper uses S3D feature extraction which seems important. Are there any other methods on STAR that use S3D? Would it be possible to run other baselines?

Next, I think this paper could be improved if it presented a method that could be learned across video datasets. The current method, at least to the best of this reviewer's understanding, has to be trained fresh on each one. As a result it might be overfitting to some of the dataset statistics of STAR/AGQA/EgoTaskQA and as a result absorbing some amount of human task-specific supervision from those sources.

Finally it woud be great to not just consider STAR/AGQA/EgoTaskQA but also other video QA datasets like MSRVTT-QA, TVQA, etc.

nit: L137-138: RoBERTa doesn't use WordPiece, it uses BPE tokenization

**Questions:**

What do you see the role of each of the losses? Are all important, and could any be removed? Are there any architecture ablations that could be done while keeping the model size the same?

**Limitations:**

nope, would be great to add such a section.

---

> ### Author Rebuttal · Authors · 2023-08-06
>
> We thank all the suggestions for a more comprehensive evaluation of our model.
> **Ablation Study On the Loss Functions**
>
> Due to space limitations, we provide a complete ablation study in the Supplementary Materials. According to the reviewer's suggestion, we try to give a more detailed analysis of the loss functions.
>
> We carefully evaluate all pairs of all losses on all three selected datasets. The results are shown in Tab.1.
>
> **Table 1. Ablation study of the loss functions on all datasets.**
>
> | Uns/Sup | $L_{cls}^{uns}$ | $L_{iou}$    | $L_{cert}$   | $L_{cls}^{sup}$ | $L_{L1}$     | Acc on AGQA v2 | Acc on STAR (Test) | Acc on EgoTaskQA |
> | ------- | --------------- | ------------ | ------------ | --------------- | ------------ | -------------- | ------------------ | ---------------- |
> | Uns     | $\checkmark$    | $\checkmark$ |              |                 |              | 54.12          | 53.62              | 42.69            |
> | Uns     | $\checkmark$    |              | $\checkmark$ |                 |              | 52.89          | 51.35              | 41.77            |
> | Uns     |                 | $\checkmark$ | $\checkmark$ |                 |              | 54.27          | 53.72              | 42.85            |
> | Uns     | $\checkmark$    | $\checkmark$ | $\checkmark$ |                 |              | **55.08**      | **53.86**          | **43.06**        |
> | Sup     |                 |              |              | $\checkmark$    |              | 54.26          | 53.79              | 43.87            |
> | Sup     |                 |              |              |                 | $\checkmark$ | 53.19          | 53.03              | 42.64            |
> | Sup     |                 |              |              | $\checkmark$    | $\checkmark$ | **55.08**      | **53.94**          | **44.27**        |
>
> - Firstly, the model with complete losses achieves the best results on both unsupervised and supervised settings of all datasets. It indicates that each loss plays its role in generating the event memories.
> - Secondly, under the unsupervised setting, the $L_{iou}$ shows an important role in controlling the temporal prediction of the events (2nd Row) while the losses of $L_{cls}$ and $L_{cert}$ seem not much impact when removing one of them. For supervised setting, the $L_{cls}$ shows more import role than the $L_{L1}$.
>
> Thus, the conclusion is that temporal constraint is more important in an unsupervised manner. Without label information, the temporal prior is much more reliable. On the contrary, label supervision is more impact in supervised learning, since the event with explicit semantics has stronger grounding ability. Therefore, one can only contain the import losses for simplification.
>
> **Ablation Study On the Modules**
>
> To the author's understanding, we try our best to keep the model parameters consistent. The Glance-only we used in the paper only replace the multi-level attention with the standard cross-attention. They have same parameters. See discussions with **Reviewer xLg6** about the **Effectiveness Analysis of the Focusing Chain** according to Tab.1in [Rebuttal](https://openreview.net/forum?id=J6Niv3yrMq&noteId=dPUKw93v1V). Besides, as the reviewer suggested, we also verify the effectiveness of the glance module (Action Detection-Focus). See detailed analysis in the [Rebuttal](https://openreview.net/forum?id=J6Niv3yrMq&noteId=t7xsOOjmQ1) with **Reviewer 16Tf**.
>
> **Table 2. Ablation study of each module on the STAR dataset.**
>
> | Model                  | Acc       |
> | ---------------------- | --------- |
> | MDETR                  | 46.26     |
> | Glance-only            | 53.12     |
> | Action Detection-Focus | 49.85     |
> | Glance-Focus           | **53.94** |
>
> **Ablation Study on Video Backbone**
>
> The current SOTA methods evaluated on STAR datasets use different backbones. As the reviewer suggested, we try different video backbones and evaluated them on the STAR dataset.
>
> **Table 3. Ablation study of video backbone on STAR dataset.**
>
> | Model                                         | Visual Backbone | Val Acc          |
> | :-------------------------------------------- | --------------- | :--------------- |
> | [ClipBERT](https://arxiv.org/abs/2102.06183)  | ResNet50        | -                |
> | [CLIP](https://arxiv.org/abs/2103.00020)      | CLIP            | -                |
> | [RESERVE](https://arxiv.org/abs/2201.02639)   | ViT-B/32        | -                |
> | [AIO](https://arxiv.org/abs/2203.07303)       | ViT-B/32        | 47.54            |
> | [MIST-AIO](https://arxiv.org/abs/2212.09522)  | ViT-B/32        | 49.69            |
> | [MIST-CLIP](https://arxiv.org/abs/2212.09522) | CLIP            | *51.13*          |
> | **GF(uns)-S3D**                               | S3D             | **53.86(+2.73)** |
> | GF(uns)-CLIP                                  | CLIP            | 52.75(+1.62)     |
> | GF(uns)-ViT                                   | ViT-B/32        | 50.94(+1.25)     |
>
> Our GF with different visual backbones achieves SOTA performances compared to the methods using the same backbones. Therefore, the framework is effective with different backbones.
> **Generalization of the Model**
>
> Current VideoQA models except those large VL models all suffer from the issues of dataset transfer gap for the answer vocabularies are various. However, GF has the potential to solve various video understanding tasks, discussed in the [Rebuttal](https://openreview.net/forum?id=J6Niv3yrMq&noteId=t7xsOOjmQ1). Therefore it can be extended to a VL pre-training model with stronger event reasoning ability.
>
> What's more, we evaluate two other VideoQA datasets in the [Global Rebuttal](https://openreview.net/forum?id=J6Niv3yrMq&noteId=LQPJMIxMFG), which shows the generalization of our model.
>
> *See limitation discussions in the [Rebuttal](https://openreview.net/forum?id=J6Niv3yrMq&noteId=ePRr6BdSrO).*

---

> > ### Comment · Reviewer_yVqG · 2023-08-11
> > **thanks for the response! keeping my score**
> >
> > thanks for the helpful response and for running these experiments! I'm still pretty confused about this paper (as i was with my original review) - about what loss functions are effective here, what backbones/etc are most effective, and how much to trust the evaluation as it might be overfitting to dataset specific statistics. for this reason, I'd like to keep my score(3) - thanks!

---

> > > ### Author Response · Authors · 2023-08-16
> > >
> > > We feel regret that our rebuttal failed to fully resolve all the concerns of Reviewer yVqG. Given the opportunity, we'd like to try again to summarize the experimental conclusions from our original [rebuttal](https://openreview.net/forum?id=J6Niv3yrMq&noteId=7Pgl2QPPl8) as follows. We sincerely hope such clarifications can offer further help.
> > >
> > > - **Loss Functions.** The loss functions we designed are shown in Tab.1 below. Only two loss functions are used in the supervised setting, $L_{L1}$ for temporal loss, $L_{cls}^{sup}$ for semantic loss. Due to the lack of event-level annotations in most cases, we **allow** unsupervised learning by **replacing** $L_{L1}$ with $L_{iou}$, **replacing** $L_{cls}^{sup}$ with $L_{cls}^{uns}$ & $L_{cert}$.
> > >
> > >   **Table 1. Loss Functions in Glance-Focus**
> > >
> > >   |              | Temporal  | Semantic                     |
> > >   | ------------ | --------- | ---------------------------- |
> > >   | Supervised   | $L_{L1}$  | $L_{cls}^{sup}$              |
> > >   | Unsupervised | $L_{iou}$ | $L_{cls}^{uns}$ & $L_{cert}$ |
> > >
> > >   Referring to [Tab.1](https://openreview.net/forum?id=J6Niv3yrMq&noteId=7Pgl2QPPl8) in the original rebuttal, both temporal and semantic constraints are necessary. And in the supervised setting, the semantic constraint is more important, while in the unsupervised setting temporal constraint is more important. One can **flexibly** use these loss functions by adjusting their weighting coefficients.
> > >
> > > - **Backbones.** For fair comparisons, we have tried various backbones which are widely used in existing comparative methods. From [Tab.2](https://openreview.net/forum?id=J6Niv3yrMq&noteId=LQPJMIxMFG) and [Tab.3](https://openreview.net/forum?id=J6Niv3yrMq&noteId=7Pgl2QPPl8) in the original rebuttal, our method has consistently achieved SOTA results in both NExT-QA and STAR datasets with the same backbones as the compared methods. About what backbones are most effective, there is no consistent conclusion, and we deem it is somewhat less relevant to our proposed method. CLIP>S3D>C3D on NExT-QA. S3D>CLIP>ViT on STAR.
> > >
> > > - **Overfitting.** We thank the reviewer's insightful suggestion. Since our model can achieve SOTA performances across all **five**  VideoQA benchmarks with various video lengths, different event numbers, diverse QA pairs, etc. We have quite strong confidence that our method has the potential for cross-dataset learning and will not overfit to a certain data distribution. We will further investigate this issue in our future study.
> > >
> > > - **Module Ablations.** Extensive experiments have been done in [Tab.1](https://openreview.net/forum?id=J6Niv3yrMq&noteId=hp0wWg88SQ) and [Tab.3](https://openreview.net/forum?id=J6Niv3yrMq&noteId=t7xsOOjmQ1) in the original rebuttal. We believe the effectiveness of the proposed glance module and focus strategy has been fully verified with **consistent** parameters.

---

### Official Review · Reviewer_sLiX · 2023-07-01

**Soundness:** 3 good
**Presentation:** 2 fair
**Contribution:** 3 good
**Rating:** 7
**Confidence:** 4

**Summary:**

The paper proposes a novel approach for multi-event video question answering (VideoQA) that mimics the human reasoning strategy of glancing and focusing. The proposed method consists of two stages: glancing and focusing. In the glancing stage, the model generates a set of episode memories that summarize the key events in the video and supports both unsupervised and supervised methods, depending on the availability of event-level annotations. In the focusing stage, it uses the episode memories as memory prompts to guide the multi-level cross-attention between questions and videos, and predicts the answer based on the focused video frames. The paper evaluates the model on three challenging multi-event VideoQA benchmarks: STAR, AGQA, and EgoTaskQA. The experimental results show that it achieves state-of-the-art performance on all datasets, and outperforms current large vision-language models in various complex reasoning tasks. The paper also conducts ablation studies and visualizations to demonstrate the effectiveness of the proposed modules and memory prompts.

**Strengths:**

1. The proposed method simulates the reasoning process exhibited by humans, namely an initial overview followed by a concentrated examination, thus presenting an interesting and novel approach.
2. The implementation is flexible and practical, enabling the simultaneous processing of annotated and unannotated videos without dependence on predetermined action vocabularies or event-level annotations.
3. This approach represents the pioneering effort to incorporate information maximization as a guiding principle for the training of unlabeled data within the realm of VideoQA.
4. It designs a multi-level cross-attention mechanism that leverages the episode memories as prompts to guide the model to focus on the relevant video frames for answer prediction.
5. The performance is quite impressive, as it even surpasses current large vision-language models in various scenarios.

**Weaknesses:**

1. The writing has a large room to be improved.
2. In L166, the proposed method assumes there are C events in a video. However, it remains unclear whether C keeps consistent across all videos or dynamically varies for every single one. If C is predetermined, it would be valuable to conduct an ablative study to assess the impact of different choices for C on the overall performance of the method.
3. For unsupervised event generation, the authors impose a strong bias that the category labels should be evenly distributed across all memories. Further elaboration is essential to explain its reasonability. As from a subjective standpoint, only a limited subset of categories can manifest in a video given the contextual constraints (e.g., playing football on the ocean is impossible).

**Questions:**

please refer to weaknesses.

**Limitations:**

The limitations have been well discussed in supplementary materials. The potential negative societal impact is recommended to be addressed as well.

---

> ### Author Rebuttal · Authors · 2023-08-10
>
> *We will certainly improve our writing in the revised version.*
>
> We thank for the reviewer motivating us to make a detailed analysis of how the event categories influence the model's performance.
>
> **Comparisons of the overall Number of Event Categories Among Datasets**
>
>  We first compare the overall event categories numbers among various datasets in the [Global Rebuttal](https://openreview.net/forum?id=J6Niv3yrMq&noteId=LQPJMIxMFG). Since the source videos are collected from different scenarios, the richness of video events in different datasets can vary from 50 to 793. But on average, there are approximately 100 common categories of events, such as those used in the STAR and AGQA datasets. However, some datasets like EgoTaskQA follow the event label annotations from LEMMA. It uses fine-grained labels which are the composition of a verb, target, etc. part of the events, therefore resulting in approximately 800 categories.
>
> In summary, the event tag itself is not easy to define, so considering this, we also propose an unsupervised setting that does not rely on action detection.
>
> **Ablation Study of the "C"**
>
> $C$ is a predetermined hyper-parameter in our model, denoting the number of event classes that may occur in the video. By default, we set it to the ground-truth overall event categories number of the dataset. As suggested by the reviewer, we conduct an ablation study to evaluate the impact of different choices for C. We separately set $C=[1,50,100,GT]$ to evaluate. The mean accuracies of the Test split on the STAR dataset are reported.
>
>  **Table 1. Ablation study on the number of event categories.**
>
> | C                 | Acc      |
> | :---------------- | --------- |
> | 1                 | 51.68     |
> | 50                | **54.58** |
> | 100               | 53.15     |
> | # GT classes(157) | *53.94*   |
>
> Set $C=1$ will severely affect the performance of the model, while setting the $C$ to a random reasonable number will only slightly affect the performance of the model. Thus, the model is not sensitive to the number of categories, which is a strength to handle the scenarios that the dataset with no event-level annotations.
>
> **Strong Bias Issue**
>
> Most interestingly, the model with $C$ set to GT number only achieves the second best result and the model with $C=50$ obtains the best result. That may somehow confirm the insightful opinion of the reviewer. The bias that the events may happen evenly in each video is too strong. Setting a smaller number will be a better choice.
>
> Besides, taking the contextual constraints into consideration may help to design a more reasonable way to generate event memories. For example, sequentially predicting each part of a compositional event, e.g., "Verb(Put)--Target(Apple)--Prep(On)--Location(Table)" could be a better way. Such a study will be done in the future.
>
> **Negative Societal Impact**
>
> Since the model is trained on human-annotated datasets, therefore the model may potentially suffer from data bias. It may cause some social problems. But overall, the potential negative social impact of this work is limited.

---

> > ### Comment · Reviewer_sLiX · 2023-08-15
> >
> > All of my concerns have been addressed. I raise my score to 7.

---

> > > ### Author Response · Authors · 2023-08-16
> > >
> > > Thank you for raising the score! Your suggestion is very enlightening to us. We are very glad that our further analysis can address all your concerns. We will accordingly revise our next version to clarify such issues.

---

### Official Review · Reviewer_xLg6 · 2023-07-05

**Soundness:** 2 fair
**Presentation:** 2 fair
**Contribution:** 2 fair
**Rating:** 6
**Confidence:** 3

**Summary:**

This paper proposes a two-stage method of constructing memories of periodical summarization of the video, then using memory prompts to local key frames for reasoning.

**Strengths:**

1. The SOTA results on three Ego QA datasets.
2. The hierarchical construction of video abstraction might be a new way of tackling lengthy videos.
3. The IoU based un-unsupervised learning could be extended to other senarios.

**Weaknesses:**

The writing needs improvement. Sentences are generally long and hard to understand.
E.g., typo in abstract "abtain".  Long sentence such as line 13-17.


The idea of one-to-one matching of discrete events with memories seem to be outdated. Why not use soft-attention to assign soft weights of each event to the memory. This seems to be a well-studied technique.


The dependency in a)-c) focus is not justified. Would this chain of dependency bring in heavy loss if M_enc is not accurate. The un-supervised construction of heuristic M seems to be not a suitable prompt. What would (9) be Q=M_foc + L_enc? Any ablation about this.

The results seem not convincing in terms of the proposed focus chain. Would this accuracy improvement come from the additional parameters?

Seems to be not a fair comparison with non-memory methods. Would methods like MIST. What would a baseline that does not construct M_enc explicitly but implicitly construct a M_foc do. Does the additional steps of constraining memory M from Eq2-5 harm the performance?

**Questions:**

The results seem not convincing in terms of the proposed focus chain. Would this accuracy improvement come from the additional parameters?

How is the positional embedding \phi formulated in Memory prompting? If it's from bi-partite assignment, is it trainable ?

Not clear how IoU is in (4). Can you provide more details.

---

> ### Author Rebuttal · Authors · 2023-08-10
>
> *We will certainly improve our writing in the revised version.*
>
> **One-To-One Matching & Soft-Attention**
>
> The one-to-one matching is only used under the supervised setting following the recently proposed series of detection works [DETR](https://arxiv.org/abs/2005.12872)/[MDETR](https://arxiv.org/abs/2104.12763)/[Moment-DETR](https://arxiv.org/pdf/2107.09609.pdf)/[Tube-DETR](https://arxiv.org/abs/2203.16434) etc. The authors take it as a simple implementation of set matching. Many other set-matching strategies are well-studied. The soft attention suggested by the reviewer can also solve this problem by training an attention weight, which may somewhat involve additional parameters. But we still looking forward to any better strategies.
>
> **Analysis of the Focusing Chain**
>
> Thanks for motivating us to study on the rationality and effectiveness of our proposed multi-level cross-attention. In our implementation, we use standard cross-attention in Transformer at each level of attention. To the authors' understanding, each level of cross attention is a weighted summary of the value $V$ (focusing) with the weight ($W^QQ^TW^KK$) reflecting the relationships between query $Q$ and key $K$. Therefore, it can realize our purpose of gradually focusing on the related visual clues from the video.
>
> **Ablation study of the Focusing Strategy**
>
> As the reviewer suggested, we ablate the strategy by reserving the question encoding when focusing on the frame. It seems like a good strategy to handle the inaccurate $M_{enc}$ issue. The accuracies on Test split of STAR dataset are shown in Tab.1. However, the performance of the suggested strategy is slightly weaker.
>
> The reasons may become the reserved question encoding may cause mixed encoding for the model, which may harm the original focus strategy. Nevertheless, before the cross-attention layer, the Decoder has self-attention layers to aggregate information across different inputs. Thus the $M_{enc}$ generated by the Glance chain is relatively effective in both unsupervised and supervised settings.
>
> **Table 1. Ablation study of the focusing strategy on STAR.**
>
> | Method                | #Parameters(MB)       | Uns       | Sup       |
> | --------------------- | --------------------- | --------- | --------- |
> | $Q=X_{foc}$(Original) | 141.427618            | **53.86** | **53.94** |
> | $Q=X_{foc}+L_{enc}$   | 141.427618            | 53.27     | 53.77     |
> | Glance-only           | 141.423522(-0.004096) | -         | 53.12     |
> | Focus-only            | 141.427618            | 52.89     | -         |
>
> **Effectiveness Analysis of the Focusing Chain**
>
> The focusing strategy will not involve additional parameters. Here, we give a detailed implementation of the Glance-only we used. Specifically, after the glance module generates the memory prompts, we concatenate them with video features as an overall visual embedding and use the standard cross-attention between visual and textual embedding to output the answer predictions. Therefore, the parameters are almost consistent. See parameters comparisons in Tab.1.
>
> Therefore, the performance gain mainly comes from our designed focusing strategy. Due to the strategy being simple but efficient, the accuracy improvement seems relatively weak.
>
> **More Baseline Implementations**
>
> As the reviewer suggested, we also implement baseline (Focus-only) with no constraints on $M_{enc}$, but keep the focusing chain remained. The results show that our designed unsupervised loss functions depend on the quality of the generated event memories.
>
> **QAs**
>
> About the positional embedding $P_t$, it is calculated by a learnable function $\Phi(.)$. About the temporal IoU loss, please see detailed explanations in [Rebuttal](https://openreview.net/forum?id=J6Niv3yrMq&noteId=C3DDfogLKK) with **Reviewer izc9**.

---

> ### Comment · Reviewer_xLg6 · 2023-08-16
> **score raised**
>
> All of my concerns have been addressed. I raise my score to 6.

---

> > ### Author Response · Authors · 2023-08-17
> >
> > Thank you for raising the score! We are glad that our response has resolved all your concerns. The relevant analyses and experimental results will be updated in the revised version.

---

### Official Review · Reviewer_izc9 · 2023-07-05

**Soundness:** 3 good
**Presentation:** 3 good
**Contribution:** 3 good
**Rating:** 7
**Confidence:** 5

**Summary:**

This paper aims to tackle the problem of multi-event VideoQA that involves multiple human-object interaction events. Inspired by the human's reasoning ability, this paper proposes the Glance-Focus model that can reason from coarse to fine. At the glancing stage, the Glance-Focus model can generate some dynamic key event memories to summarize the key events in the video. At the focusing stage, the Glance-Focus model can reason based on the most relevant moment through the cross-attention mechanism. The contributions of this paper are summarized as following:
(1) This paper proposes a glance-and-focus mechanism to reason multi-event VideoQA in a human-like manner.
(2) The Glance-Focus model achieves the state-of-the-art performance on some multi-event VideoQA dataset.


**Strengths:**

(1) The idea of glancing and focusing on multi-event VideoQA is interesting and intuitive. This human-like design brings great potential for solving such multi-event VideoQA problem. The glancing can be seen as a kind of information compression to anchor key events, which may guide the focusing stage to conduct the fine-grained reasoning.

(2) The motivation of unsupervised event generation at the glancing stage is clear. Because of the open-vocabulary world, it is hard to annotate all the events in the long-time video. To tackle this problem, this paper proposes to enforce the unsupervised constraints to generate meaningful key events.

(3) To generate useful event memory in an unsupervised manner, this paper proposes to enforce the individual discrimination and the global diversity, which can guarantee to learn more meaningful key events.

(4) This paper innovatively proposes to use event memories as anchors to prompt the model to locate the most relevant segments corresponding to the questions. Besides, the multi-level cross-attention mechanism can promote the event memory to focus on the question-related visual clues in a fine-grained manner.

(5) Extensive experiments and analyses show the effectiveness and reasonability of the method. Specifically, the visualization of Figure 4 clearly shows the relationship between the event memory generation and the fine-grained focusing.

**Weaknesses:**

(1) At the glancing stage, this paper utilizes the unsupervised learning solution to generate meaningful event memories. However, this paper lacks of the analyses about the inherent relations between the unsupervised learning (such as instance discriminative learning) and the key event memory generation.

(2) The formulation of the temporal IoU loss (Eq.(4)) is not very clear and it should be explained more detailedly.

(3) In experiments, some unsupervised results even outperform the supervised manner but necessary analyses about this phenomenon are lacking.

**Questions:**

(1) The Glance-Focus model can not only tackle the problem of multi-event VideoQA but also tackle other kinds of reasoning problems. What about its scalability? The authors should provide more analyses.

(2) In the unsupervised manner at the glancing stage, how to ensure the sequential order of events?

**Limitations:**

Yes, the authors have discussed the limitations in the supplementary material.

---

> ### Author Rebuttal · Authors · 2023-08-09
>
> We are very glad to see all the positive reviews you given. Hope we can address your questions below.
>
> **Relations with unsupervised learning**
>
> We will add more relevant explanations in the revised version. Similar to individual discriminative learning, we aim to generate a set of event-level individuals with strong discrimination. The unsupervised individual discriminative learning usually sets constraints on the pairwise sample features to learn discriminative instance features representations [[1]](https://arxiv.org/pdf/1805.01978.pdf)[[2]](https://arxiv.org/abs/1904.03436). For instance, maximize the probability of the augmented instance $\hat{x}_i$ being classified as instance $x_i$:
>
> $$
> P(x_i|\hat{x}_i)=\frac{\exp(f_i^T\hat{f}_j/\tau)}{\sum\exp(f_k^T\hat{f}_i/\tau)}.
> $$
> Therefore, these methods need to construct such sample pairs, which is inefficient. Besides, the quality of the constructed pairs will impact influence the discrimination of the representations.
>
> Instead of focusing on the feature space, we subtly utilize the prior knowledge of the event memories, i.e., the events happening in the video are usually different, different events should have small temporal overlaps, etc. According to them, we directly set constraints on their classification distribution and the predicted timestamps. Among them, we use information maximization to design the individual certainty loss $L_{cert}$ and semantic diversity loss $L_{cls}$ in Eq.(2) and (3).
>
> **Details of Event Timestamp Prediction**
>
> For the temporal IoU loss, we sum all normalized temporal IoU of each pair of events in a video. The normalized temporal IoU is denoted as:
> $$
> tIoU(\tau_i, \tau_j)= \frac{\max(0,  \max(t^b_i, t^b_j)-\min(t^e_i, t^e_j)}{\min(t^b_i, t^b_j)-\max(t^e_i, t^e_j)))},
> $$
> where the $[t^b,t^e]$ is the timestamp, i.e., the beginning and end time of the event converted from the temporal prediction $\tau$.
>
> Since we can predict the timestamp of each event, we can arrange these events in their sequential order. Without class labels, we cannot do one-to-one matching in a supervised manner, but we can ensure events happened in sequential order.
>
> **Unsupervised>Supervised Analysis**
>
> We try to explain such an interesting phenomenon.
>
> - Firstly, we think such a phenomenon is reasonable for we did not give very strong memory supervision on the model. The GF model is trained with the combined memory loss and QA loss in Eq.(14). And we set the coefficient of memory loss smaller than QA loss, for QA is the final task we want to handle.
> - Secondly, we think such a phenomenon is acceptable since we do not expect event memory strictly grounding to the event annotations. As our human beings generate abstract memories of the video, we expect dynamically adjustable event memories. Therefore, the memories with weaker supervision may become more adaptable in the reasoning stage.
>
> **Scalability to Other Video Understanding Tasks**
>
> As discussed in [Rebuttal](https://openreview.net/forum?id=J6Niv3yrMq&noteId=zdm3xQm46a) with **Reviewer 16Tf**, our GF model is easy to extend to other video understanding tasks, such as video captioning/dense captioning/summarization. We will evaluate these tasks in future work. Besides, the Glance chain of our model can be used as a strong action detection model as well, which is evaluated in this [Rebuttal](https://openreview.net/forum?id=J6Niv3yrMq&noteId=t7xsOOjmQ1).

---

> > ### Comment · Reviewer_izc9 · 2023-08-15
> > **Clear response and a solid work, keep the original rating for accpeting.**
> >
> > Thanks for the authors' clear response! The motivation of unsupervised event memory generation is clear. It is enlightening to use information maximization to generate meaningful memories in this task. The explanation about the case of "Unsupervised>Supervised" sounds reasonable and I suggest the authors that they should add these detailed analyses in their next version. Lastly, it is meaningful that GF model shows comparable performance with other action detection models. I suggest the authors that they should further extend this idea to other video understanding tasks. Based on the above analyses and other reviewers' postive comments, I keep the original rating to accept this maniscript.

---

> > > ### Author Response · Authors · 2023-08-16
> > >
> > > We‘d like to thank you again for your affirmation of our work. We are very glad that our response can address all your concerns. We will incorporate the detailed analyses and extended results in the revised version and we are trying to extend our work to the video captioning task. If you have any further questions or concerns, welcome to discuss with us. Your feedback is greatly appreciated!

---

### Official Review · Reviewer_16Tf · 2023-07-06

**Soundness:** 3 good
**Presentation:** 3 good
**Contribution:** 2 fair
**Rating:** 6
**Confidence:** 4

**Summary:**

The paper introduces a model called Glance-Focus for Video Question Answering (VideoQA) tasks, which involve understanding human behaviors in videos. While vision language models have shown success in multi-modal tasks, reasoning over long videos with multiple human-object interaction events remains challenging. Humans can tackle this task effectively by using memory to quickly locate key moments for reasoning. To mimic this reasoning strategy, the Glance-Focus model is proposed.

At the glancing stage, an Encoder-Decoder is trained to generate dynamic key event memories by aggregating relevant video frames. This is achieved through unsupervised memory generation using individual discriminating and global diversity loss, as well as supervised bipartite matching losses. At the focusing stage, the event memories enable the model to focus on the key event memory and then on the most relevant moment for specific reasoning using multi-level cross-attention.

The model is evaluated on three representative VideoQA benchmarks (STAR, EgoTaskQA, and AGQA) and achieves state-of-the-art results, surpassing current large models in challenging tasks. The Glance-Focus model addresses the challenge of situation reasoning in VideoQA by generating dynamic event memories and using them to focus on key moments in videos.

**Strengths:**

The paper's organization is good, it clearly explains the basic ideas.

The paper proposed to involve memory to cross the gap between the concepts and video content in the QA transformers.

The related work is organized well and will be better to introduce the recent benchmarks together with the VideoQA.

The implemented experiments are sufficient and the example visualization is helpful.

**Weaknesses:**

Firstly, in Table 1, AIO / Temp[ATP] / MIST results appear to be directly cited from [1] but seem to be based on the STAR validation part rather than the test part. Only RESERVE is reported on the STAR test. The dataset partition used for evaluation must be clarified for a meaningful comparison.

In Section 4.1, please specify the average video length of EgoTaskQA. If available, additional statistics, such as the average number of events/action state changes per dataset, would be beneficial since the proposed method detects state changes in the 'glance' stage. This data will aid in understanding how the method handles scenes with varying event complexity.

Furthermore, the paper lacks qualitative evaluation metrics for action/event/moment detection, such as IoU/mAP. These are essential for analyzing accumulated errors.

The paper asserts the impracticality of using an action detector/event detector due to the close-set results it produces. However, a qualitative comparison with the proposed method using event/action detectors + rule-based filtering (keywords matching), e.g. Moment-DETR [2] and TubeDETR [3], is absent.

Lastly, although the paper discusses multi-event detection, it still chooses multiple keyframes/visual clues in the focusing stages, which has been explored in previous works [4] and [5, 6]. A more comprehensive discussion comparing these related works would be appreciated.

Some typos: Line 233 HowTo100m → HowTo100M.


[1] D. Gao et al, MIST: Multi-modal Iterative Spatial-Temporal Transformer for Long-form Video Question Answering
[2] J. Lei et al, Detecting moments and highlights in videos via natural language queries
[3] A, Yang et al, TubeDETR: Spatio-Temporal Video Grounding With Transformers
[4] S. Yu et al, Self-Chained Image-Language Model for Video Localization and Question Answering
[5] S. Kim, Semi-parametric video-grounded text generation
[6] T. Qian et al, Locate before answering: Answer guided question localization for video question answering

**Questions:**

Refer to the above sections.

**Limitations:**

The codes should be open-source but it seems the paper doesn't mention

---

> ### Author Rebuttal · Authors · 2023-08-09
>
> **Corrections on STAR Dataset**
>
> We are very grateful for the reviewer's reminder. The following are the updated results, with GF in the unsupervised setting achieving SOTA on both the Test and Val parts.
>
> **Table 1. Corrections on STAR evaluations.**
>
> | Model                                         | Test Acc  | Val Acc          |
> | --------------------------------------------- | --------- | ---------------- |
> | [ClipBERT](https://arxiv.org/abs/2102.06183)  | 36.7      | -                |
> | [CLIP](https://arxiv.org/abs/2103.00020)      | 38.0      | -                |
> | [RESERVE](https://arxiv.org/abs/2201.02639)   | 40.5      | -                |
> | [Flamingo](https://arxiv.org/abs/2204.14198)  | 43.4      | 42.23            |
> | [AIO](https://arxiv.org/abs/2203.07303)       | -         | 47.54            |
> | [Temp[ATP]](https://arxiv.org/abs/2206.01720) | -         | 48.37            |
> | [MIST](https://arxiv.org/abs/2212.09522)      | -         | *51.13*          |
> | **GF(uns)**                                   | **57.82** | **53.86(+2.73)** |
>
> **Detailed Dataset Analysis**
>
> We really appreciate the suggestions. The in-depth analysis of the datasets is presented in [global rebuttal](https://openreview.net/forum?id=J6Niv3yrMq&noteId=LQPJMIxMFG). Our proposed method performs well on datasets with various average event numbers per video from 2.7(STAR) to 8.8(NExT-QA). Compared to current VideoQA models, the proposed Glance-Focus frameworks can better handle complex multi-event video understanding scenarios. Besides, the method also exhibits robust performance on videos covering different action categories from 50(NExT-QA) to 793(EgoTaskQA). See also in [Rebuttal]() with **Reviewer sLiX**.
>
> **Effectiveness Analysis of the Glance Stage**
>
> Since our proposed event memory is an implicit representation of videos, it is difficult to analyze the accuracy of event generation, especially in the absence of event annotation. Therefore, only quantitative visualization experiments were conducted (Fig. 3/4 in the main paper, Fig. 2 in the supplementary materials).
>
> Based on the suggestions of the reviewer, we further conduct event accuracy evaluations on the STAR/AGQA. We evaluate on the [Charades](https://arxiv.org/abs/1604.01753) dataset, which is the video source of these two datasets, and use the standard [action detection setting](https://prior.allenai.org/projects/data/charades). For a fair comparison, we report the prediction of the Glance Chain from GF trained with the same action supervision. The results are shown in Tab. 2. GF model shows comparable performance with other methods, which indicates that the Glance chain of our model can effectively capture events (actions) in the video.
>
> **Table 2. Action Detection Comparisons with SOTA Methods on Charades Benchmark**
>
> | Method                                                       | mAP  |
> | ------------------------------------------------------------ | ---- |
> | [Coarse-Fine](https://arxiv.org/abs/2103.01302) | 25.1 |
> | [PDAN](https://paperswithcode.com/paper/pdan-pyramid-dilated-attention-network-for) | 26.5 |
> | GF(sup)                                                      | 23.7 |
>
> **Ablation Study of Using Action Detection Model**
>
> Thanks for motivating us to conduct such a necessary ablation study. As the reviewer suggested, we try to use [SlowFast](https://arxiv.org/abs/1812.03982), a popular action detection model, to replace our Glance chain and handle the open-set issue by keyword mapping. [Moment-DETR](https://arxiv.org/pdf/2107.09609.pdf) and [Tube-DETR](https://arxiv.org/abs/2203.16434) are more related to our work. However, they aim at a video grounding task with a textual query, which is a different task from action detection/classification. To alleviate training an additional action detection model, we straightly use the SlowFast model trained on the [AVA](https://arxiv.org/pdf/1705.08421.pdf) dataset. It can detect actions within a 60-category close-set vocabulary. Then, we map these actions to the actions annotated in the STAR dataset (157 classes overall) by keyword matching. Then, the filtered actions are used as memories and inputted into the Focus chain for QA.
>
> **Table 3. Ablations on the Glance chain. The results are mean accuracies on the STAR Test Split.**
>
> | Method                 | Acc   |
> | :--------------------- | ----- |
> | Action Detection-Focus | 49.85 |
> | Glance-Focus(sup)      | 53.94 |
>
> The performance of such a strategy is not that good. The reasons come from the action category gap across different datasets. As stated in the paper, the events in different scenarios are various and difficult to define at the same time. See more discussions on [Rebuttal](https://openreview.net/forum?id=J6Niv3yrMq&noteId=ucQstYUtv5) with the **Reviewer sLiX**. Therefore, unsupervised event generation is necessary, and our work provides an effective implementation.
>
> **Comparison with Related Works**
>
> Thank you for providing the latest related work. We will update the comparison with them in the revision. Similarly, we all utilize the idea of finding the most related video content for QA. But all these works directly use questions as queries to retrieve keyframes, while GF dynamically finds visual clues with memory prompts. Compared to them, GF has the following advantages.
>
> - **More accurate.** Keyframes retrieval suffers from the semantic gap between questions and frames. Memory can bridge the gap for they are middle-level representations.
> - **More adaptable.** GF can extract dynamic and adjustable visual cues across the whole video with the multi-level attention, while the above works can only find clues based on the fixed keyframes.
> - **Extensibility.** GF can obtain a complete compact representation of videos without relying on question queries, so it can be extended to more video comprehension tasks, such as video captioning/dense captioning/summarization.
>
> *P.S. The code will be made public after the paper is accepted.*

---

> > ### Comment · Reviewer_16Tf · 2023-08-18
> > **Feedback**
> >
> > The responses answered most of the questions and provided sufficient experiments. Thank you for your efforts.

---

> > > ### Author Response · Authors · 2023-08-19
> > >
> > > We sincerely appreciate your positive support of our work after reviewing our rebuttal. Thanks for your thorough consideration of our responses and give affirmation of our efforts. If you have any further questions or concerns, please don't hesitate to discuss them with us.

---

### Official Review · Reviewer_veHn · 2023-07-07

**Soundness:** 3 good
**Presentation:** 3 good
**Contribution:** 3 good
**Rating:** 6
**Confidence:** 4

**Summary:**

This paper addresses the task of question-answering in videos which have multiple events. The authors draw inspiration from the human capability to create event memories as a way to anchor the reasoning process over multiple events, and propose the Glance-Focus model, which aggregates a set of event memories (during a first "glancing" stage) and then proceeds to further process these memories ("focus" stage) to produce the final answer to the query. The authors benchmark their model on several standard multi-event benchmarks, such as STAR, AGQA (the balanced v2 version), and EgoTaskQA to characterize the effectiveness of their approach.

**Strengths:**

`+` Question-answering in videos is an important task and challenging domain for multimodal models. Further, the goal of temporal reasoning is an important step for the development of AI agents in general.

`+` The general direction of reasoning on top of generated/matched event memories draws inspiration from human event memory/reasoning, and seems like a reasonable approach.

`+` The authors evaluation their technique on several benchmarks (representing different video distributions) and show improvements; I also appreciated that the authors looked at the v2 version of AGQA, for better characterization of the model capabilities.

**Weaknesses:**

`-` The paper emphasizes "long video" understanding, but the length of the videos in the datasets examined are generally quite short (12-30 seconds) relative to many other video QA settings (e.g., ActivityNet-QA is ~2 mins). I understand this may be an inherited term from the prior work, MIST [10], which also uses similar terminology. However, that prior work did at least evaluate on NExT-QA, which is ~50% longer on average than the longest video dataset examined in this work here (AGQA).

`-` Additionally, while the results on the short benchmark (STAR) are significant, the results become significantly more mixed when compared with closely related work (e.g., [10]) on longer benchmarks like AGQAv2 (+0.69 overall, and some specific categories like "sequencing", which seems like it should be better with event reasoning, are much worse than prior work). This also brings to question the efficacy of the proposed approach on longer video settings.

`-` Further, the mixed results are even more striking due to the fact that this work leverages stronger low-level event supervision (for the bipartite event memory matching) than prior work (e.g., MIST) does. It is unclear how, without this level of supervision, the gains of the proposed technique, relative to prior work, are impacted (or the limitations of this approach as the videos become longer).

**Questions:**

Overall, the rating for this paper pre-rebuttal is borderline. If the authors can address and clarify the points/questions raised in the weaknesses above, that would be helpful for updating the final rating and better grounding the discussion phase with the other reviewers.

**Limitations:**

The authors don't seem to provide a detailed discussion of limitations in the work, though the societal impacts does seem to be similar to related work in this space. It would be helpful and clarifying if there was an better indicator of how aspects of the proposed technique, e.g. the supervised bipartite matching, scale with longer videos, and how such limitations may be better addressed by future work.

---

**Post-rebuttal update:** The authors largely address some of the key concerns that were brought up in the original review, in particular, the additional analysis on NExT-QA significantly improves the evaluation (the results on ActivityNet are promising, but not the *primary* focus of the updated review given the context and discussion below). While there are some still areas that can still be further improved (as detailed and described in the rebuttal discussion below), I am increasing my rating towards leaning acceptance since I believe that these are areas that can be further addressed by future work.

---

> ### Author Rebuttal · Authors · 2023-08-08
>
> **Scale to longer video**
>
> We thank the reviewer's suggestions. The evaluations on longer VideoQA benchmarks (NExT-QA/ActivityNet-QA) are shown in the [global rebuttal](https://openreview.net/forum?id=J6Niv3yrMq&noteId=LQPJMIxMFG). Note that the model is tested under the unsupervised setting since the two datasets both do not have event-level annotations. These results can further indicate the effectiveness of the proposed approach on longer video settings.
>
> **Event Reasoning Ability Analysis**
>
> We found that [MIST](https://arxiv.org/abs/2212.09522) performs very well on the sequencing questions compared to other works. Specifically, it achieves a 13.88% improvement compared to GF. For further analysis, we calculate the breakdown accuracy results on question types (superlative/sequencing/duration) related to multi-event reasoning ability by the answer types (binary/open-ended). The results in Tab.1 indicate that although MIST performs better than GF in question types such as sequencing and duration, its performance advantages mainly come from the binary answer questions, which may indicate that the model overfits the data bias. Nevertheless, GF performs better on all open-ended questions, which reflects its stronger generalization ability.
>
> **Table 1.  Breakdown results on question types related to event reasoning by the answer types.**
>
> | Methods                                  | Superlative                      | Sequencing                     | Duration                       |
> | ---------------------------------------- | -------------------------------- | ------------------------------ | ------------------------------ |
> | [AIO](https://arxiv.org/abs/2203.07303)  | 37.53                            | 49.61                          | 50.81                          |
> | [ATP](https://arxiv.org/abs/2206.01720)  | 39.78                            | 48.25                          | 51.79                          |
> | [MIST](https://arxiv.org/abs/2212.09522) | *42.05*(B: **56.36**/O: *28.35*)     | **67.24**(B: **67.43**/O: *26.45*) | **60.33**(B: **55.33**/O: 24.59)   |
> | **GF-obj(sup)**                          | **44.62**(B: 56.20/O: **33.62**) | *53.36*(B: 53.35/O: **26.91**) | *52.80*(B: 53.43/O: **25.61**) |
>
> Additionally, we also report the breakdown results on the longer NExT-QA dataset by the different question types(Causal/Temporal/Descriptive) in Tab.2. According to the results, GF shows a strong temporal reasoning ability than MIST. It indicates that GF still has strong event reasoning ability although the videos become longer.
>
> **Table 2. Evaluation breakdown on NExT-QA by the different question types.**
>
> | Methods                                                      | Acc@Causal | Acc@Temporal     | Acc@Descriptive  |
> | ------------------------------------------------------------ | ---------- | ---------------- | ---------------- |
> | [HGA](https://ojs.aaai.org/index.php/AAAI/article/view/6767) | 46.26      | 50.74            | 59.33            |
> | [Just Ask](https://arxiv.org/abs/2012.00451)                 | 51.4       | 49.6             | 63.1             |
> | MIST                                                         | *56.6*     | *54.6*           | *66.9*           |
> | **GF(uns)**                                                  | **56.93**  | **57.07(+3.07)** | **70.53(+3.63)** |
>
> **Limitation Discussion**
>
> We have provided discussions on limitations in the supplementary materials. Based on the all reviews, we further summarize and update the limitation discussions here.
>
> - The current videos collected in the Video QA datasets are not long enough and the event is not complex enough to fully evaluate the model's multi-event reasoning ability. We look forward to further progress on multi-event reasoning research. Also, future works including keyframe selection strategies are considered to improve the long video understanding efficiency.
>
> - The unsupervised setting is not as effective as the supervised setting. However, event annotations are expensive. In the future, more efficient and effective event-generation methods will be considered, such as automatically generating pseudo labels for unsupervised training.
>
> - The constraints on the generation of the event memory under unsupervised settings are relatively straightforward. More intuitive ideas from the way how human beings generate event memories are desired to be explored.

---

> > ### Comment · Reviewer_veHn · 2023-08-21
> > **Thank you for the rebuttal**
> >
> > Thank you for the rebuttal! I think the updated results/detailed analysis is promising, and feel the rebuttal has addressed many of my key concerns. A couple of additional notes:
> >
> > **ActivityNet-QA results (clarifying claims and details)**
> >
> > As a clarification, my understanding is that this model is actually *not* state-of-the-art on ActivityNet-QA, counter to what is suggested in this part of the global author rebuttal:
> > > Furthermore, as Reviewer VeHn & yVqG suggested, to verify our model on more VideoQA datasets, we additionally test on longer video understanding benchmarks, ActivityNet-QA. The performance shows that GF also achieves SOTA result and even surpasses the Just-Ask model pre-trained on large VideoQA data.
> >
> > The main comparison with JustAsk (ICCV 2021) is helpful, but there are more relatively recent works with much higher accuracies. As one example, FrozenBILM from last year's NeurIPS 2022 (see `[M1]` below), which achieves `43.2` (see Table 6, "w/o speech"). I think the provided results are still promising, but *the claims should be made more accurate/calibrated* (right now, the statement seems like it is unequivocally the SOTA across all prior work, rather than perhaps an improvement on some specific prior model/feature setting in prior work).
> >
> > Related to the above, I think I have a general idea of what settings were used based on prior work, but if the authors can provide a *confirmation* of details on what were the specific features/model/task protocol/etc details, that would be helpful too. For example, specifying the relation to the JustAsk baseline, which used frozen S3D-HowTo100M features (which I assume is likely the same as ones used in this work, per L232-233 in the original submission, but possibly some other features like CLIP also?) + some specific protocols.
> >
> > `[M1]` Yang et al. "Zero-Shot Video Question Answering via Frozen Bidirectional Language Models." NeurIPS 2022.
> >
> > **AGQAv2/NExT-QA results (additional general comments)**
> >
> > The AGQAv2 analysis is helpful, and better illustrates the difference between this method and MIST. This analysis, combined with the NExT-QA experiments, really help to flesh out the core empirical support for the work. The NExT-QA results are promising, and it is good to see both the overall results and the detailed breakdown on C/T/D compared with MIST (especially, showing a modest improvement on C, and significant one on T). As an additional suggestion from recent/concurrent work (for a later revision, not necessary for now), it would be interesting to see if the performance improvement deltas increase further when compared on the "hard" C/T subset from [3].
> >
> > **Overall:** I appreciate the effort the authors put for the rebuttal, and feel it has made an overall net positive impression on the work (of course, assuming the claims/details/etc will be clarified in a later revision). I will plan to update my final review and rating during the final reviewer-AC discussion phase.

---

> > > ### Author Response · Authors · 2023-08-21
> > >
> > > We are glad that our response can address your concerns. We will update the additional experimental results, detailed experiment settings in the later revision, and modify the claims in the experimental analyses. Explanations on your additional notes are as follows.
> > >
> > > **ActivityNet-QA results**
> > >
> > > Thanks for the reviewer's kind reminder. We will revise our statement to make our claim more precise. Considering that ActivityNet-QA does not belong to the typical multi-event video QA benchmark, we only showed a concise experimental analysis for the limited space.
> > >
> > > We further explain the details of the experimental setup under this dataset. From the observation in Tab.2 of our [Global Rebuttal](https://openreview.net/forum?id=J6Niv3yrMq&noteId=LQPJMIxMFG) on the NExT-QA benchmark, GF model with CLIP feature is better than C3D/S3D feature. So we exploit CLIP feature on the more challenging ActivityNet-QA with longer videos, by using only QA labels as supervised information from the train split of the dataset and evaluating on its test split. The Just Ask baseline used the frozen S3D-HowTo100M video features. The specific comparison is shown in Tab.1. As for your mentioned recent work FrozenBiLM, it is yet another model pre-trained on the large dataset WebVid10M, which is consist of 10 million video-text pairs and thus significantly benefits the final accuracy.
> > >
> > > **Table 1. Detailed comparision on ActivityNet-QA.**
> > >
> > > | Method                                                       | Pre-training Data | Training Data  | Features | Acc on ActivityNet-QA |
> > > | ------------------------------------------------------------ | ----------------- | -------------- | -------- | --------------------- |
> > > | [HGA](https://ojs.aaai.org/index.php/AAAI/article/view/6767) | -                 | ActivityNet-QA | C3D      | 34.6                  |
> > > | [LocAns](https://arxiv.org/abs/2210.02081)                   | -                 | ActivityNet-QA | C3D      | 36.1                  |
> > > | [Just Ask](https://arxiv.org/abs/2012.00451) (w/o pre-trained) | -                 | ActivityNet-QA | S3D      | 36.8                  |
> > > | GF-CLIP(uns)                                                 | -                 | ActivityNet-QA | CLIP     | **41.1**              |
> > > | [Just Ask] (pre-trained)   | HowToVQA69M       | ActivityNet-QA | S3D      | 38.9                  |
> > > | [FrozenBiLM](https://arxiv.org/abs/2206.08155)               | WebVid10M         | ActivityNet-QA | CLIP     | 43.2                  |
> > >
> > > Considering the fairness of the comparison, we did not compare the performance with more large pre-trained VL models. Similar to FrozenBiLM, these models all use additional large-scale data to train the networks. By designing different pre-training tasks, this type of models can achieve very good results on single-event video QA benchmarks, such as ActivityNet-QA. For more results, please refer to the [leaderboard](https://paperswithcode.com/sota/video-question-answering-on-activitynet-qa) of ActivityNet-QA. On the contrary, we focus on proposing a strong multi-event video QA model that does not rely on large-scale pre-training data. Nevertheless, as discussed in "Generalization of the Model" from the [rebuttal](https://openreview.net/forum?id=J6Niv3yrMq&noteId=7Pgl2QPPl8), our method can also be extended to a pre-training model based on video-text pairs, which will be considered in our future work.
> > >
> > > **NExT-QA results**
> > >
> > > We are glad that our analysis on NExT-QA dataset is helpful. As the reviewer suggested, we tried our best to further evaluate our model on the ATP-hard subset of this dataset in this last day of the Author-Reviewer discussion phase. Due to the extremely limited time, we are only allowed to make a brief comparison with the methods that have reported the results on this subset in their original papers. A more complete comparison with methods such as MIST will be updated in our revision. From Tab.2, our model is shown to still achieve strong performance on both hard Causal and Temporal questions compared to models with the same video features. Among the comparative methods, CoVGT demonstrates superior accuracy, by additionally involving the fine-grained object-level modeling with Faster R-CNN video features.
> > >
> > > **Table 2. Evaluation results on ATP-hard subset of NExT-QA.**
> > >
> > > | Methods                                   | Features          | Acc@C    | Acc@T    |
> > > | ----------------------------------------- | ----------------- | -------- | -------- |
> > > | ATP                                       | CLIP              | 19.6     | 22.6     |
> > > | Temp[ATP]                                 | CLIP              | 38.4     | 36.5     |
> > > | HGA                                       | C3D               | 43.3     | 45.3     |
> > > | [CoVGT](https://arxiv.org/abs/2302.13668) | CLIP+Faster R-CNN | **51.8** | 50.5     |
> > > | GF-CLIP(uns)                              | CLIP              | 48.7     | **50.8** |

---

> > > > ### Comment · Reviewer_veHn · 2023-08-21
> > > > **Thank you for the additional response**
> > > >
> > > > Thank you for the detailed follow-up message! Once again, I appreciate the efforts of the authors.
> > > > - On ActivityNet, the additional details are helpful for contextualizing the results (e.g., use of CLIP vs. S3D, etc.), and the points raised by the authors are reasonable (e.g., ActivityNet is a "concise experimental analysis" outside the standard multi-events benchmarks set). It would still be interesting (as part of _optional_ future work) to see how GF performs with the same frozen S3D features, since these were used in other parts of this work (and are comparable to what is used in "JustAsk, w/o pretrained"). Overall, I do think the results are promising, even if they are not the absolute state-of-the-art, and I am glad that the authors will plan to include the additional necessary context and edits to their claims in a revised paper/supplement.
> > > > - On NExT-QA, the updated evaluation is helpful and the results seem to be good, especially considering extra object features were not used for GF. I understand the additional comparison with MIST will take some time for the reasons mentioned, and in particular, will require more time than the scope of the rebuttal period -- this was expected and why I mentioned "(for a later revision, not necessary for now)" in my earlier response. Nonetheless, seeing the preliminary result now in a short time frame is a pleasant surprise and is also helpful.
> > > >
> > > > I'm still planning to finalize and update my review/rating during the final discussion, but in case it was not clear earlier, I do plan to increase my rating based on the author rebuttal.

---

> ### Author Response · Authors · 2023-08-20
> **Kindly Reminder -- we are expecting the reviewer's reply to our rebuttal, thanks!**
>
> Dear Reviewer veHn,
>
> We appreciate your relatively positive review of our work such as motivation reasonable and suppose our paper has the potential to be given a higher score. Therefore we have tried our best to address every of your raised issues. We hope these will help clarify all your concerns and we are sincerely anticipating your feedback. We’d like to give a brief summary of our answers to your questions as follows.
>
> Motivated by your comments, we have conducted a comparative analysis of the current Video QA benchmarks in Tab.1 from the [Global Rebuttal](https://openreview.net/forum?id=J6Niv3yrMq&noteId=LQPJMIxMFG). Then we augmented the evaluation of our method on the NExT-QA and ActivityNet-QA datasets, which contain much longer videos. Even if these two datasets do not contain event annotations, our method can surpass the current SOTA methods (including MIST) without such supervision, as shown in Tab.2 and Tab.3 from the [Global Rebuttal](https://openreview.net/forum?id=J6Niv3yrMq&noteId=LQPJMIxMFG). Also, we have made further analysis of the event reasoning ability of our model on the AGQA v2 dataset as suggested in Tab.1 from the [Rebuttal](https://openreview.net/forum?id=J6Niv3yrMq&noteId=ePRr6BdSrO). The performance advantages of MIST mainly come from the binary answer questions which may show the overfitting phenomenon. However, our method demonstrates better results on the more difficult open-ended split. What's more, from the breakdown evaluation on the NExT-QA dataset in Tab.2 of the [Rebuttal](https://openreview.net/forum?id=J6Niv3yrMq&noteId=ePRr6BdSrO), our model also has strong event reasoning ability when scaling to longer videos.
>
> Warm regards,
>
> The Authors

---

### Author Rebuttal · Authors · 2023-08-08

**We thank all reviewers for the insightful feedback. We are encouraged that you find our motivation interesting and reasonable. We try to address the common issues below.**

P.S. For convenience, we directly link the paper of the related works. The best results in the tables are **bolded**, and the second-best results are in *italics*. We may cross reference to the related rebuttal.

**Selection and Analysis of the Evaluation Datasets**

We first clarify the reason for selecting the current popular evaluation benchmarks. In this paper, we mainly focus on challenging Multi-Event Video QA tasks. Motivated by **Reviewer 16Tf & yVqG**, we provide a further analysis of the current VideoQA benchmarks through comparisons on the number of events per video, average video length, and overall action classes number in Tab.1. These factors mainly come from the original dataset statistic. Numbers with * are estimated results since no annotations are provided. We comprehensively consider these factors to choose appropriate evaluation datasets for model evaluation in the main paper.

**Table 1. Datasets Analysis.**

| Datasets                                                    | # Avg. Events/ Video | Avg. Length | # Event Classes |
| :---------------------------------------------------------- | :-----------: | :---------: | --------------- |
| [MSVD-QA](https://dl.acm.org/doi/10.1145/3123266.3123427)   |      1*       |     10s     | 100*            |
| [MSRVTT-QA](https://dl.acm.org/doi/10.1145/3123266.3123427) |      1*       |     15s     | 100*            |
| [ActivityNet-QA](https://arxiv.org/abs/1906.02467)          |      1.4      |    180s     | 203             |
| [STAR](https://openreview.net/pdf?id=EfgNF5-ZAjM)           |      2.7      |     12s     | 157             |
| [AGQA](https://arxiv.org/pdf/2103.16002.pdf)                |      6.8      |     30s     | 157             |
| [EgoTaskQA](https://arxiv.org/pdf/2210.03929.pdf)           |       5       |     25s     | 793       |
| [NExT-QA](https://arxiv.org/abs/2105.08276)                 |      8.8      |     44s     | 50              |

From the table, we can find that MSVD-QA/MSRVTT-QA/ActivityNet-QA only contains about one event.  Therefore they mainly focus on single-event understanding. Note that **longer videos may not necessarily contain complex content**. Though the video lengths are relatively longer in ActivityNet-QA, they only contain one event or multiple repetitions of a single activity. The video contents in STAR/AGQA/EgoTaskQA/NExT-QA are more complex since they have multiple events happening in the video. In addition, the richness of event categories varies among different datasets. According to the statistics, there are approximately 100 common categories of events on average.

This paper mainly focuses on Multi-Event Video QA task, with the challenge of temporal and causal reasoning among multiple events. Therefore, STAR/AGQA/EgoTaskQA are selected for evaluation. Due to the lack of event-level annotation in the NExT-QA dataset, we cannot conduct experiments under supervised settings in the main paper. However, as **Reviewer VeHn** suggested, the experiments under unsupervised settings on the NExT-QA dataset have been supplemented.

**Table 2 Evaluation results on NExT-QA.**

| Methods                                       | Val Acc          | Test Acc         |
| --------------------------------------------- | ---------------- | ---------------- |
| [Just Ask](https://arxiv.org/abs/2012.00451)  | 52.3             | -                |
| [IGV](https://arxiv.org/abs/2206.02349)       | -                | 51.34            |
| [ATP](https://arxiv.org/abs/2206.01720)       | 54.3             | -                |
| [VGT](https://arxiv.org/pdf/2207.05342.pdf)   | 55.02            | 53.68            |
| [HGA+EIGV](https://arxiv.org/abs/2207.12783)  | -                | *53.7*           |
| [MIST-CLIP](https://arxiv.org/abs/2212.09522) | *57.18*          | -                |
| GF-C3D(uns)                                   | 56.33            | 54.75            |
| GF-S3D(uns)                                   | 56.44            | 55.60            |
| **GF-CLIP(uns)**                              | **58.83(+1.53)** | **57.92(+4.22)** |

We evaluate the Glance-Focus model (GF for short) on the NExT-QA dataset based on three different backbones. The GF models with the same backbone as other methods achieve SOTA results. Such results further reflect the strong multi-event reasoning ability of our method, for NExT-QA is the most complex Multi-Event VideoQA benchmark according to the analysis above.

Furthermore, as **Reviewer VeHn & yVqG** suggested, to verify our model on more VideoQA datasets, we additionally test on longer video understanding benchmarks, ActivityNet-QA. The performance shows that GF also achieves SOTA result and even surpasses the Just-Ask model pre-trained on large VideoQA data.

**Table 2 Evaluation results on ActivityNet-QA.**

| Methods                                                      | Acc            |
| :----------------------------------------------------------- | -------------- |
| [HGA](https://ojs.aaai.org/index.php/AAAI/article/view/6767) | 34.6           |
| [LocAns](https://arxiv.org/abs/2210.02081)                   | 36.1           |
| Just Ask                                                     | 36.8           |
| Just Ask(Pre-trained)                                        | *38.9*         |
| **GF(uns)**                                                  | **41.1(+2.2)** |

---

### Comment · Area_Chair_zaZj · 2023-08-13
**Discussion**

Dear Reviewers,

Thank you for reviewing this paper. Authors have provided their rebuttal. Would you please check it, and give your comments/rating based on the rebuttal letter and the comments from other reviewers?

Best Regards

AC

---

### Decision · Program_Chairs · 2023-09-21

**Decision:**

Accept (poster)

**Comment:**

This paper has been extensively reviewed by six experts. Among those, five reviewers recommend acceptance. Overall, most of the reviewers think the method is interesting and reasonable, e.g., reviewer izc9: "The idea of glancing and focusing on multi-event VideoQA is interesting and intuitive", reviewer vehn: "a reasonable approach"; reviewer sLiX: "The proposed method simulates the reasoning process exhibited by humans, thus presenting an interesting and novel approach". However, reviewers do identify some limitations of this work, which includes the limitations in clarity of the writing, detailed analysis of experiments and comparisons with some works. Authors have addressed most of the questions raised by the reviewers. As for the concerns in terms of ablation studies (eg. backbones) and analysis (e.g., loss functions) raised by reviewer yVqG, authors have also addressed. Considering the novelties of the work and efficacy of the method, AC recommends accept, and urges the authors to incorporate the improvements and additional experiments and analysis to the final version.